# Spatial transcriptomics landscape of lesions from non-communicable inflammatory skin diseases

A. Schäbitz [1,10], C. Hillig [2,10], M. Mubarak[3], M. Jargosch[3,4], A. Farnoud[2], E. Scala[1,5], N. Kurzen [3], A. C. Pilz[4,5], N. Bhalla [6], J. Thomas[3], M. Stahle[1], T. Biedermann [4], C. B. Schmidt-Weber [3], F. Theis [2], N. Garzorz-Stark[1,4], K. Eyerich[1,5,7,10], M. P. Menden [2,8,9,10] & S. Eyerich [3,10] ✉

Abundant heterogeneous immune cells infiltrate lesions in chronic inflammatory diseases and characterization of these cells is needed to distinguish disease-promoting from bystander immune cells. Here, we investigate the landscape of non-communicable inflammatory skin diseases (ncISD) by spatial transcriptomics resulting in a large repository of 62,000 spatially defined human cutaneous transcriptomes from 31 patients. Despite the expected immune cell infiltration, we observe rather low numbers of pathogenic disease promoting cytokine transcripts (*IFNG, IL13* and *IL17A*), i.e. >125 times less compared to the mean expression of all other genes over lesional skin sections. Nevertheless, cytokine expression is limited to lesional skin and presented in a disease-specific pattern. Leveraging a density-based spatial clustering method, we identify specific responder gene signatures in direct proximity of cytokines, and confirm that detected cytokine transcripts initiate amplification cascades of up to thousands of specific responder transcripts forming localized epidermal clusters. Thus, within the abundant and heterogeneous infiltrates of ncISD, only a low number of cytokine transcripts and their translated proteins promote disease by initiating an inflammatory amplification cascade in their local microenvironment.

Non-communicable inflammatory diseases (ncISD) are based on complex interactions of predisposing genetic background and environmental triggers that collectively result in altered immune responses. Several hundred ncISD exist, including lichen planus (LP), atopic dermatitis (AD), and psoriasis. Despite their heterogeneity, most ncISD can be categorised according to adaptive immune pathways based on the interaction of distinct lymphocyte subsets with the epithelium[1,2]. Whereas psoriasis represents a classical type 3 immune cell mediated

[1]Division of Dermatology and Venereology, Department of Medicine Solna, and Center for Molecular Medicine, Karolinska Institutet, Stockholm, Sweden. [2]Institute of Computational Biology, Helmholtz Zentrum München—German Research Centre for Environmental Health, Ingolstädter Landstrasse 1, 85764 Neuherberg, Germany. [3]Center for Allergy and Environment (ZAUM), Technical University and Helmholtz Center Munich, Biedersteinerstrasse 29, 80802 Munich, Germany. [4]Department of Dermatology and Allergy, Technical University of Munich, Biedersteinerstrasse 29, 80802 Munich, Germany. [5]Department of Dermatology and Venerology, Medical Center—University of Freiburg, Faculty of Medicine, University of Freiburg, Freiburg, Germany. [6]Department of Gene Technology, School of Engineering Sciences in Chemistry, Biotechnology and Health, KTH Royal Institute of Technology, Stockholm, Sweden. [7]Department of Dermatology and Venereology, Unit of Dermatology, Karolinska University Hospital, Stockholm, Sweden. [8]Department of Biology, Ludwig-Maximilians University, Goßhadernerstrasse 2, Martinsried 82152, Germany. [9]German Center for Diabetes Research (DZD e.V.), Ingolstädter Landstrasse 1, 85764 Neuherberg, Germany. [10]These authors contributed equally: A. Schäbitz, C. Hillig, K. Eyerich, M.P. Menden, S. Eyerich. ✉e-mail: Stefanie.Eyerich@tum.de

disease[3,4], AD is dominated by type 2[5,6], and LP by type 1[7,8] immune cells. Accordingly, psoriasis can be efficiently treated with antibodies targeting cytokines of type 3 immunity, i.e., IL-17A or IL-23[9,10]. Likewise, AD is successfully treated with antibodies targeting cytokines of type 2 immune cells, such as IL-13[11,12]. However, without models to predict therapeutic responses, many patients do not respond to a given therapy. Furthermore, we lack curative approaches, since current therapies neutralize cytokines, but do not target antigen-specificity. More granular information regarding the profile, kinetics, and spatial distribution of cytokine-secreting immune cells is needed to achieve a substantial advance in addressing these challenges.

Emerging molecular techniques allow analysis of mRNA expression in single-cell and spatial contexts, thus enabling deep phenotyping of relevant cell types in ncISD[13,14]. Conventional single-cell sequencing techniques require dissociation of the tissue and thereby might bias the interpretation due to loss of tissue context. Spatial transcriptomics (ST) overcomes this issue, allowing the study of the inflamed skin architecture[15,16], however, not on single cell resolution. Investigating disease-driving cells together with their direct responder signatures in a spatial context will offer further insights into the pathogenic microenvironment of ncISD.

In this work, we investigate adaptive immune responses in lesional and non-lesional skin of ncISD with spatial resolution using the Visium technology of 10X Genomics. We observe that single transcripts of disease-promoting cytokines, namely IFNG for LP, IL13 for AD, and, IL17A for psoriasis initiate localized amplification cascades of specific inflammatory responder genes that collectively represent hallmarks of the respective disease pathogenesis. Thus, a few immune cells promote ncISD within an abundant heterogeneous infiltrate.

## Results

To analyze the pathogenic microenvironment of non-lesional and lesional ncISD skin, we characterised the spatial transcriptomic landscape of ncISD (Fig. 1a), covering LP, AD and psoriasis. Gene expression was measured in frozen and H&E stained skin sections using the Visium technology of 10X Genomics. Here, the tissue is spatially resolved in so-called spots that are distributed equally over the whole tissue section. Each spot has a diameter of 55 μm and is distanced from the neighboring spot center-to-center by 100 μm. The generated dataset included 90 samples (31 lesional, 14 matched non-lesional in duplicates) and the transcriptomes of 62,968 spots. After removing 3649 spots with unique molecular identifier (UMI) counts below 1 and mitochondrial fraction above 25% (Methods), 15,285 non-lesional and 44,034 lesional spots entered further analyses.

We proposed two complementary analysis workflows (Fig. 1b–g) that allow insights into the spatial distribution and function of

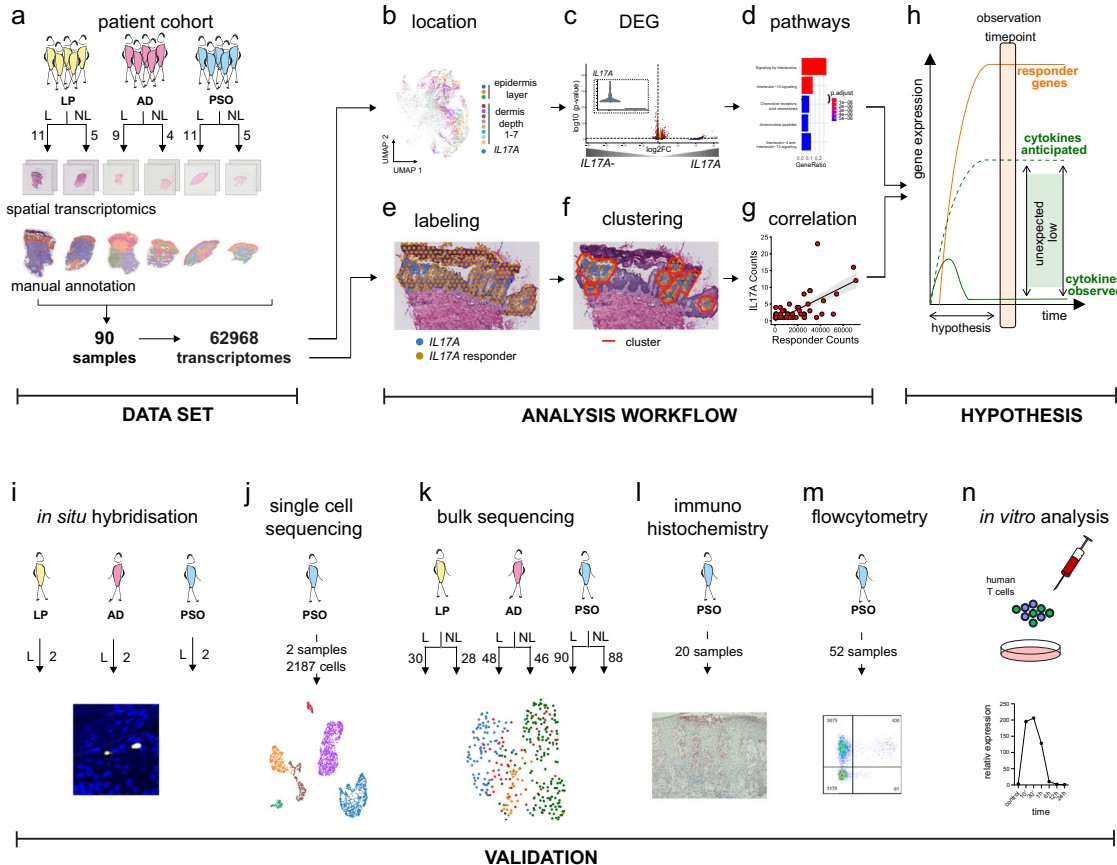

**Fig. 1 | The study design highlighting the spatial transcriptomic dataset, the analysis pipeline, and the validation cohorts and techniques. a** ST dataset consisting of 90 spatial samples (31 patients) 62 lesional samples, 28 non-lesional samples, and three ncISD (lichen planus (LP), atopic dermatitis (AD), psoriasis (Pso)) resulting in 62,968 transcriptomes. Within the analysis workflow, every spot in all samples was manually annotated according to tissue localization (basal-, middle-, upper epidermis, dermis depth 1-7). **b** Leukocyte-positive spots were defined by the expression of either *CD2, CD3D, CD3E, CD3G, CD247 (CD3Z), or PTPRC (CD45)* or combinations of markers and cytokine transcript-positive leukocytes were assigned to a tissue localization. **c** Differential gene expression (DEG) analysis was performed on cytokine transcript-positive versus cytokine transcript-negative leukocyte spots, followed by (**d**) pathway enrichment analysis. For spatial correlation of cytokine transcript-positive leukocyte spots with cytokine responder genes: (**e**) spots were labeled as cytokine or responder positive, (**f**) clusters of cytokines and responders were defined, and (**g**) correlation analysis was performed.
**h** Hypothesis expecting higher cytokine mRNA counts than observed. General low expression of cytokine transcripts in ncISD was confirmed using (**i**) in situ hybridization, (**j**) single-cell sequencing, (**k**) bulk sequencing, (**l**) immunohistochemistry, (**m**) flow cytometry, and (**n**) in vitro stimulation of human T cells.

leukocytes producing central disease cytokines in human skin. For simplification we depicted *IL17A* as a reference model in the figure panel. The first workflow (Fig. 1b–d) identified the spatial location of cytokine transcript-positive leukocyte spots (Fig. 1b, Methods) and uses these spatial features for differential gene expression (DEG) analysis of spots containing cytokine transcript-positive *versus* cytokine transcript-negative leukocytes (Fig. 1c), followed by pathway enrichment analysis (Fig. 1d). The second workflow (Fig. 1e–g) labelled cytokine transcript-positive leukocyte spots (Fig. 1e), and then used our density-based clustering method (Fig. 1f) to correlate cytokine and responder gene signatures according to spatial features (Fig. 1g). This analysis led to the surprising observation that single cytokine transcripts initiated amplification cascades of thousands of specific responder transcripts, which are causative and disease driving in the tissue microenvironment (Fig. 1h). We validated the results using a variety of patient cohorts and techniques such as in situ hybridisation, single-cell and bulk sequencing, immunohistochemistry, flow cytometry and cell culture analysis (Fig. 1i–n).

## Low numbers of disease-promoting cytokine transcripts are expressed in lesional skin of ncISD

As cytokines represent important drivers of tissue inflammation in ncISD, we examined the expression of the major effector cytokines driving the common ncISD LP, AD, and psoriasis namely *IFNG, IL13* and *IL17A*, respectively, in spatial resolution (Fig. 2a). Taking the whole section into account, we merely identified 434, 144 and 224 UMI counts for *IFNG, IL13* and *IL17A*, respectively, distributed over 372, 103 and 154 spatial spots, respectively, in all lesional sections investigated (Fig. 2e, Supplementary Fig. 1a, Supplementary Table 2). Generally, respective cytokine transcripts were unequally distributed across all samples, with AD being particularly heterogeneous (Supplementary Table 2). As expected, the number of cytokine UMI counts was low in non-lesional skin samples (Fig. 2b, c). Here, we observed mean UMI counts of 1, 1 and 0 for *IFNG, IL13*, and *IL17A*, respectively (Supplementary Table 2). However, even in lesional ncISD skin, we detected only a few cytokine transcripts ranging from 1 to 37, 1 to 12, and 1 to 27 transcripts/section for *IFNG, IL13* and *IL17A*, respectively (Fig. 2d, e, Supplementary Table 2). To exclude that cytokine-transcript positive spots were removed during quality control measures, we re-analyzed removed spots with UMI count <1 and/or mitochondrial fraction above 25% highlighting that those spots did not contain cytokine transcripts. To not thin out cytokine positivity, we included double-positive spots into our analysis, albeit them representing a minority (Supplementary Figs. 1c, 2c). In relation to the mean of all other genes, *IFNG, IL13* and *IL17A* together were 900-times and 125-times less expressed in non-lesional and lesional skin, respectively (Supplementary Fig. 1c) underlining the scarce presence of cytokines in human skin.

Despite their low frequency, the spatial distribution, however, was distinct for the investigated cytokines. While *IFNG* (basal epidermis + dermis 1 vs upper + middle epidermis + dermis 2-7 $p = 1.66e^{-22}$) and *IL13* (middle + basal epidermis + dermis 1 and dermis 2 vs upper epidermis + dermis 3-7 $p = 2.41e-17$) were significantly enriched in the lower epidermis layers and upper dermis layers, *IL17A* was detected in all layers of the lesional epidermis and was scarcely expressed in the dermis (epidermis vs dermis $p = 2.96e^{-13}$) (Fig. 2a, d, Supplementary Fig. 1a). We validated our observation of low transcript numbers and low numbers of cytokine transcript-positive spots in inflamed tissue throughout the ST dataset using various ex vivo and in vitro methods. In situ hybridization identified very few cytokine transcript-positive signals (Fig. 2f). The median number of transcript-positive cells per section for *IFNG, IL13*, and *IL17A* mRNA were 83, 4 and 11 for LP, AD, and psoriasis, respectively, thus confirming our observations from the ST analysis (Fig. 2g). In line, single-cell RNASeq analysis of psoriasis also indicated few transcripts per *IFNG* or *IL17A transcript-positive* cells, with a median UMI count for *IFNG* or *IL17A* of 1 per CD4+ cell and 4 per

CD8+ cell (Fig. 2h–j). We also investigated a large cohort of ncISD patients using bulk RNA sequencing. Here, in a third of a 6 mm skin punch biopsy we detected a median of 1 and 25.5 counts/biopsy for *IFNG* in non-lesional and lesional LP skin, respectively (Fig. 2k–m). In AD, we measured a median of 2 and 4.5 counts/biopsy of *IL13* and in psoriasis a median of 0 and 7.5 *IL17A* counts/biopsy in non-lesional and lesional skin, respectively. Immunohistochemistry and flow cytometric analysis of skin-infiltrating T cells showed comparable numbers of cytokine-positive lymphocytes in lesional skin (histology: 13.3% IL-17A+ lymphocytes, flow cytometry: 4.2% CD4+IL-17A+, 4.9% CD8+IL-17A+ (Supplementary Fig. 2a–c)). Time course analysis showed that short T cell receptor (TCR) stimulation in vitro resulted in transient mRNA production with a peak at 10–30 min and a total production time of <6 h. Low numbers of mRNA transcripts per cell increased with prolonged TCR stimulation (Supplementary Fig. 2d).

Despite their low UMI counts, cytokines showed a disease-specific expression pattern in spatial resolution. *IFNG* transcripts were mostly expressed in lesional LP (median/section:4), *IL13* (median/section:1.5) and *IL17A* (median/section:9) in AD and psoriasis, respectively, (Fig. 2n–p, Supplementary Fig. 3b) with an emphasized expression in upper skin layers (Fig. 3a, b, Supplementary Fig. 3a, c). The distinct distribution pattern held true for other disease-driving cytokines such as *IL17F, IL21, IL22, TNF, IL10*, and *IL4* (Supplementary Fig. 1d). The relative distribution of the signature cytokines confirmed that LP is a type 1, AD a type 2, and psoriasis a type 3 immune-driven disease (Fig. 2q, Supplementary Fig. 1d, e). Taken together, these findings show that low numbers of disease-specific cytokine transcripts are present in inflamed skin and show a characteristic tissue distribution.

## Cytokine transcript-positive spots and nearby spots are characterized by specific gene expression signatures

To phenotype these cytokine transcript-positive spots, we performed DEG analysis of cytokine transcript-positive *versus* cytokine-transcript-negative spots. To specifically focus on immune cells, spots were pre-sorted according to leukocyte markers (*CD2, CD3D, CD3E, CD3G, CD247* (*CD3Z*), or *PTPRC* (*CD45*)). Presence of at least one UMI count of a single or combination of these markers was regarded as a leukocyte-positive spot. Due to the size of every spot (Ø55μm), DEG generally displayed genes derived from cytokine-producing cells and genes originated from cells that respond to the given cytokine in close proximity. Given that a spot with a diameter of 55 μM can contain several cells, we used Tangram[17] to deconvolute and predict the cellular composition of cytokine transcript positive leukocyte spots generating predictive spatial maps of cell types in a given spot. Here, Tangram showed varying levels of T cells and innate immune cells as main producers of cytokines in representative ST sections (Supplementary Fig. 4a–i). In line with this, *IFNG transcript*-positive spots were characterized by genes related to type 1 immune cells, such as *GZMB, FASLG, CD70, CXCR3*, and *CXCR6*, and by genes that are induced by *IFNG* in epithelial cells such as *CXCL9, CXCL10 and CXCL11*(Supplementary Fig. 5a). *IL13 transcript*-positive spots presented themselves with differentially expressed signatures being associated with type 2 cells, such as *IL2, IL10*, and *SLAMF1*, plus genes associated with their tissue response, among them *CCL17, CCL22, MMP12*, and *OSM* (Supplementary Fig. 5c). Genes associated with *IL17A transcript-positive spots* were *IL17F, IL22*, and *IL26*, and genes being induced by *IL17A in the skin* e.g., *IL19, NOS2, S100A7A, DEFB4A, CXCL8*, and *IL36G* (Fig. 3c, d). This strength of ST in locating immune cell derived genes together with their correlating tissue response was further illustrated by a gene set enrichment analysis of lead cytokine transcript-positive spots, showing specific signatures for both inflammation-driven cell signaling and tissue reaction to inflammation (Fig. 3e, Supplementary Fig. 5b, d).

Using our psoriasis scRNA-seq dataset, unsupervised clustering identified distinct cell types (Fig. 4a), and disease-driving cytokines were almost exclusively detected in the lymphocyte cell cluster

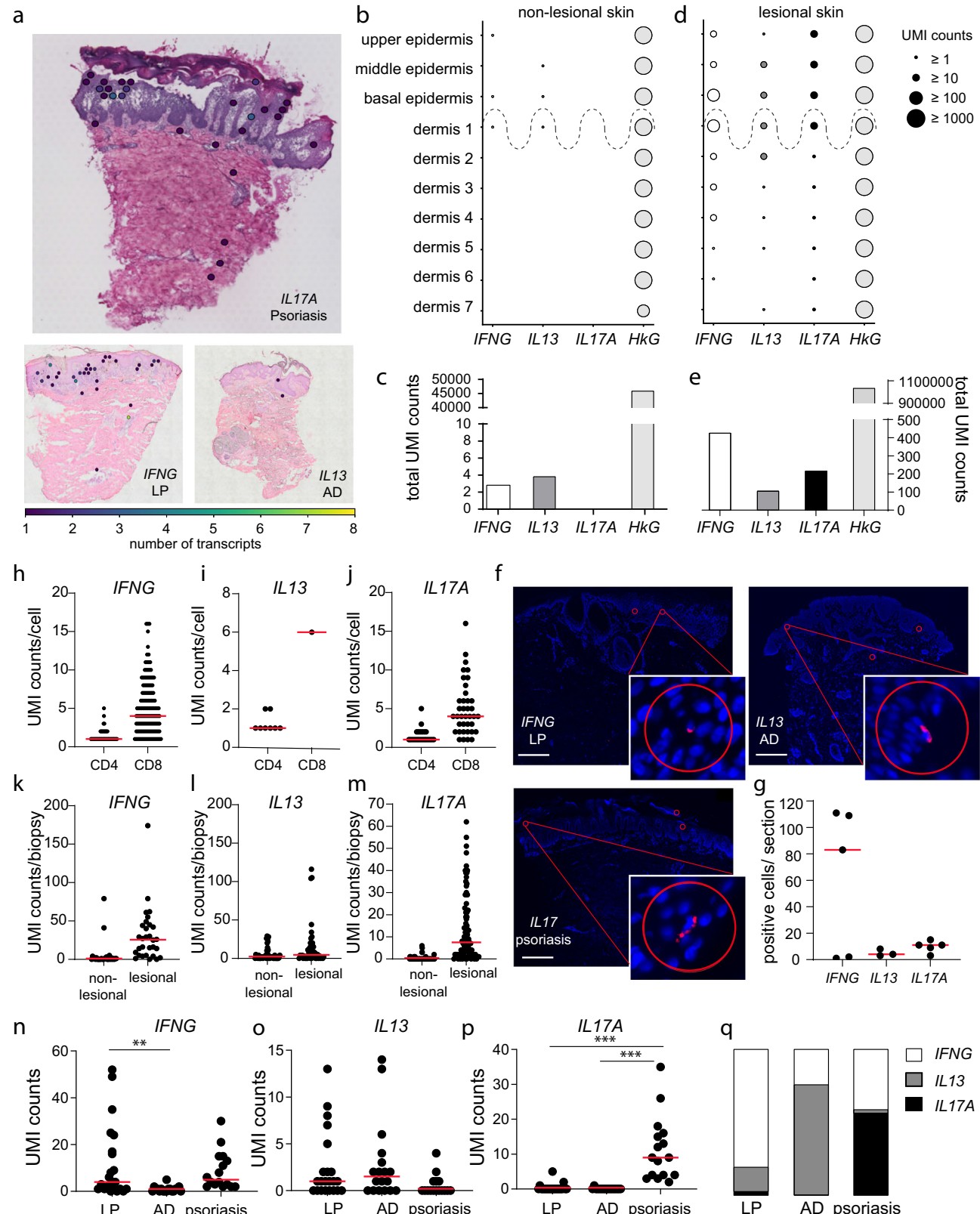

(Fig. 4b, Supplementary Fig. 6a). DEG analysis of cytokine transcript-positive *versus* cytokine transcript-negative leukocytes further confirmed the spatially determined expression of immune cell derived genes on single-cell level. However, response signatures that cytokines induce in tissue cells were widely missing in the single-cell leukocyte fraction as these genes were either not detected or found below statistical significance (Fig. 4c). This comparison, however, also

highlighted that 2 genes (*CCL4* and *CCL5*) being induced by *IFNG* in epithelial cells could also be detected in *IFNG* transcript-positive leukocytes of the single-cell dataset (Supplementary Fig. 6b) and thereby pointed out that these chemokines can be produced also by leukocytes.

In essence, we identified gene signatures that define cytokine transcript-positive spots in lesional skin and highlighted that spatial

**Fig. 2 | Low numbers of disease-driving cytokine transcripts are expressed in lesional skin of ncISD. a** Representative ST sections for psoriasis with *IL17A*-, AD with *IL13*, and LP with *IFNG transcript-positive* spots (Ø55μM). **b, d** UMI-counts of *IFNG* (white), *IL13* (dark grey) and *IL17A* (black) expressed in the manually annotated tissue layers 'upper, middle, and basal epidermis' and 'dermis depth 1-7' in non-lesional (**b**) and lesional skin (**d**) of all investigated samples (*n* = 82). *GAPDH* serves as a housekeeping gene (HkG) (light grey). **c, e** Total cytokine (*IFNG* (white), *IL13* (dark grey) and *IL17A* (black)) and *GAPDH* (light grey) UMI counts in all non-lesional (**c**) and lesional (**e**) skin sections. **f** In situ hybridization for *IFNG*, *IL13* and *IL17A* in representative stainings of LP (upper left panel), AD (upper right panel) and psoriasis (lower panel). Scale bar indicates 500 μm; red circles represent the size of a spot (Ø 55 μm) and indicate the positivity for cytokine mRNA. **g** Quantification of cytokine-positive cells per in situ section. Given are *IFNG transcript*-positive cells in

LP (*n* = 5), *IL13 transcript*-positive cells in AD (*n* = 3) and *IL17A transcript*-positive cells in psoriasis (*n* = 5). **h–j** scRNA-seq analysis of psoriasis biopsies (*n* = 2, 2187 cells) indicating the UMI count of *IFNG* (178 cells), *IL13* (9 cells), and *IL17A* (61 cells) per cell in CD4 or CD8 co-expressing cells. **k–m** Bulk sequencing analysis of non-lesional and lesional LP (*n* = 30) (*IFNG*), AD (*n* = 48) (*IL13*), and psoriasis (*n* = 90) (*IL17A*) biopsies indicating the total UMI counts for *IFNG*, *IL13*, and *IL17A*, respectively, in each biopsy. **n–p** UMI counts for *IFNG*, *IL13*, and *IL17A* in ST sections separated by disease (each dot represents one section) (LP *n* = 22, AD *n* = 18, Pso *n* = 18). Statistical significance was determined using One-Way Anova and Turkey's multiple comparisons test without FDR correction. **<0.01, ***<0.001. **q** Percentage of disease relevant cytokines in LP, AD, and psoriasis normalised to 100%. LP lichen planus, AD atopic dermatitis.

resolution allows us to understand not only immune cell derived genes, but also the response they induce in close spatial proximity in the inflammatory microenvironment.

## Immune response is spatially correlated with cytokine transcript number

To further investigate the functional relevance of the observed few cytokine transcripts in lesional ncISD skin, we studied the correlation between cytokine transcripts and their induced response in surrounding spots (Fig. 1e–g). To verify a specific response signature for each cytokine and given the fact that cytokines were mostly expressed in the epidermis, we stimulated primary human keratinocytes in vitro with recombinant IFN-γ, IL-13 or IL-17A and performed gene expression arrays to retrieve differentially expressed genes (DEG) for each cytokine (Supplementary Fig. 7a). After filtering these DEGs for log2FC and *p*-value, the gene list was compared to the spatially derived DEG lists of each cytokine (Fig. 3c, Supplementary Fig. 5a, c) delivering a specific response signature for IFN-γ (29 genes), IL-13 (4 genes plus 10 literature derived genes) and IL-17A (21 genes) (Supplementary Fig. 7b–d). The determined responder genes were equally distributed overall the dataset (Supplementary Fig. 7e–g) and 270 times more expressed than *IFNG, IL13* and *IL17A* together (Supplementary Fig. 7h). Initially, these responder gene counts were correlated with their matching cytokine counts in all epidermal ST sections without taking the spatial resolution into account (Fig. 5A–C). *IFNG* had a strong correlation with its responders (weighted Spearman $r = 0.62$; $p = 3.54e^{-10}$), whereas *IL13* and *IL17A* had low positive correlations with the respective responder genes (weighted Spearman $r = 0.39$; $p = 3.42e^{-4}$ and $r = 0.22$; $p = 4.74e^{-2}$, respectively). We were next interested if the spatial information would improve the correlation between cytokine- and responder transcripts. We therefore developed a density-based clustering method that uses spatial information. By correlating the presence of a cytokine transcript with its responder genes in the same spot (radius 0) or neighbouring spots (radius 1-9) we could identify distinct radiuses of action for each cytokine (Fig. 5D, E). Whereas *IL17A* showed its strongest effect in the direct surrounding (radius 0), effects of *IFNG* were quite stable across all radiuses investigated with a peak at radius 4 (Fig. 5E). *IL13's* action peaked at radius 3 and declined at higher radiuses (Fig. 5E). We then used the identified optimal radius of action for each cytokine and leveraged our density cluster method to investigate the correlation of cytokine- and responder transcripts. Density-based clustering markedly improved the correlation between cytokines and epidermal tissue response in the inflammatory microenvironment for *IFNG* (weighted Spearman $r = 0.73$; $p = 1.5e^{-10}$), *IL13* (weighted Spearman $r = 0.57$; $p = 1.3e^{-3}$), and *IL17A* (Pearson $r = 0.83$; $p = 9.13e^{-21}$), (Fig. 5F–H). Strikingly, the few cytokine-positive spots having only 1–15 (*IFNG*: 1 to 8, *IL13*: 1 to 3, *IL17A*: 1 to 15 UMI counts/spot) cytokine transcripts were able to induce up to 25,000 responder transcripts in the surrounding spots indicating a tremendous amplification of the cytokine signal and thereby an amplification of tissue inflammation.

We furthermore wondered if the density-based clustering method could be challenged to identify new cytokine-specific response genes. For this, we performed a DEG analysis comparing cytokine transcript-positive spots with all remaining spots in the epidermis, which were not part of any cytokine cluster of the respective classified radius of action. With a log2FC cut-off of >1 and padj-value < 0.05, we thereby identified 974 *IFNG-related*, 148 *IL13-related, and 228 IL17A-related upregulated DEGs* (Supplementary Fig. 8a–c). By this, we data-driven expanded our definition of cytokine-gene associations such as *SRGN, LYZ* and *CCL17, CLEC10A and GM2A* for *IFNG, IL13 and IL17A*, respectively (Supplementary Fig. 8).

In summary, regions with more cytokine transcripts had higher response signatures compared to regions with no or less cytokine transcripts. Consequently, the inclusion of the spatial information and density-based clustering enhanced the biological signal for all cytokines and their response signatures. Altogether, these results provide comprehensive insights into the relationship between cytokine transcript-expressing cells and their induced tissue response and confirm our hypothesis that a low number of transcripts is sufficient to induce pathogenic immune responses in the skin.

## Discussion

Curative therapies of common inflammatory skin diseases have seemed unrealistic as these diseases typically show an infiltrate of abundant and heterogeneous immune cells into lesional skin. However, new molecular techniques and bioinformatic tools allow us to dissect ncISD on a new level and to undertake first steps in the development of curative therapies. Here, we investigated ncISD with spatial resolution. Namely, we explored the molecular landscape using DEG analysis and developed an algorithm to investigate the impact of cytokine transcripts on their direct surrounding environment. We demonstrated that a minority of immune cells actively drive the pathology of ncISD by producing low numbers of signature cytokine transcripts. Indeed, these few cytokine transcripts then translate into thousand-fold higher induction of pro-inflammatory response genes, thus inducing an amplification cascade forming an inflammatory microenvironment and subsequently leading to tissue damage and pathology. Our analysis is largely based on ST generating a unique dataset of both lesional and non-lesional skin samples of patients suffering from inflammatory skin diseases. Preserving spatial information, while being independent of long digestion steps, is enormously beneficial in tissue systems like skin with distinguishable functional units. In essence, ST enables researchers to investigate whole transcriptome sequencing data in the context of interacting units in complex tissues. However, it is clear that ST is subjected to methodical challenges. Further refinement needs to be implemented in terms of the spatial resolution that with a distance from the neighboring spot center-to-center of 100 μm is omitting valuable information and with a spot diameter of 55 μm captures more than just one cell. So far it cannot be ruled out that presence of one cell type does affect the activity and response of another cell type. Being a technology with

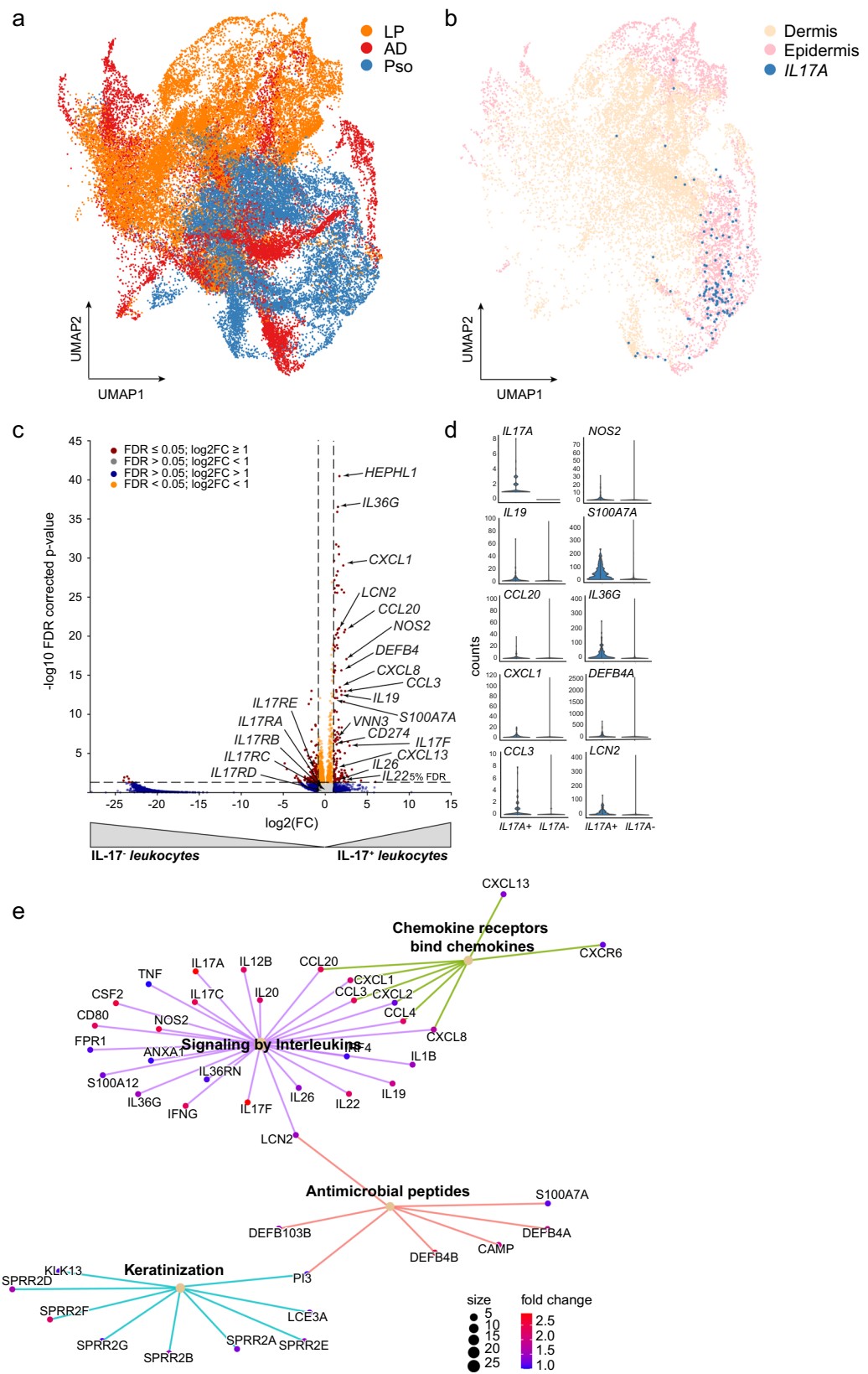

extended resolution properties, we additionally performed in situ hybridisation to support our ST analysis and, by delivering comparable results, confirmed our central findings. To deconvolve spatial spots, we moreover applied the state-of-the-art Tangram algorithm[17], which is a heuristic method to computationally predict the cellular composition of every spot of interest. Even though further reduction of the

spot diameter is highly desirable and may well be the next evolutionary step in ST, our analysis shows that exploiting the capabilities of ST to the fullest offers unique research opportunities and empowers to investigate the architecture of skin inflammation.

Despite cytokine transcripts being rare in inflamed skin, they were detectable in disease- and spatial-specific patterns. The distribution

**Fig. 3 | IL17A transcript-positive leukocyte spots are characterized by Th17 markers and *IL17A* tissue response genes. a** UMAP (Uniform Manifold Approximation and Projection) plot showing the distribution of spots within the analyzed diseases AD, LP and Pso and non-lesional (NL) skin. **b** ST spots expressing *IL17A* transcripts and leukocyte marker genes and their location in epidermis or dermis. IL17A-positive spots are highlighted in blue. **c** Volcano plot analysing the gene expression profile of *IL17A transcript*-positive leukocyte (*IL17A*⁺) *versus IL17*

transcript-negative leukocyte (*IL17A*⁻) spots. Coordinates for *IL17A* (38.4/168.3) are not shown. Benjamini–Hochberg was used to determine statistical significance. **d** Violin plots of selected genes in *IL17A* transcript-positive leukocyte (*IL17A*⁺) and *IL-17A* transcript-negative leukocyte (*IL17A*⁻) spots indicate the expression of gold standard genes. **e** Pathway enrichment analysis of genes co-expressed with *IL17A* in spatial spots. LP lichen planus, AD atopic dermatitis, Pso psoriasis, ST spatial transcriptomics.

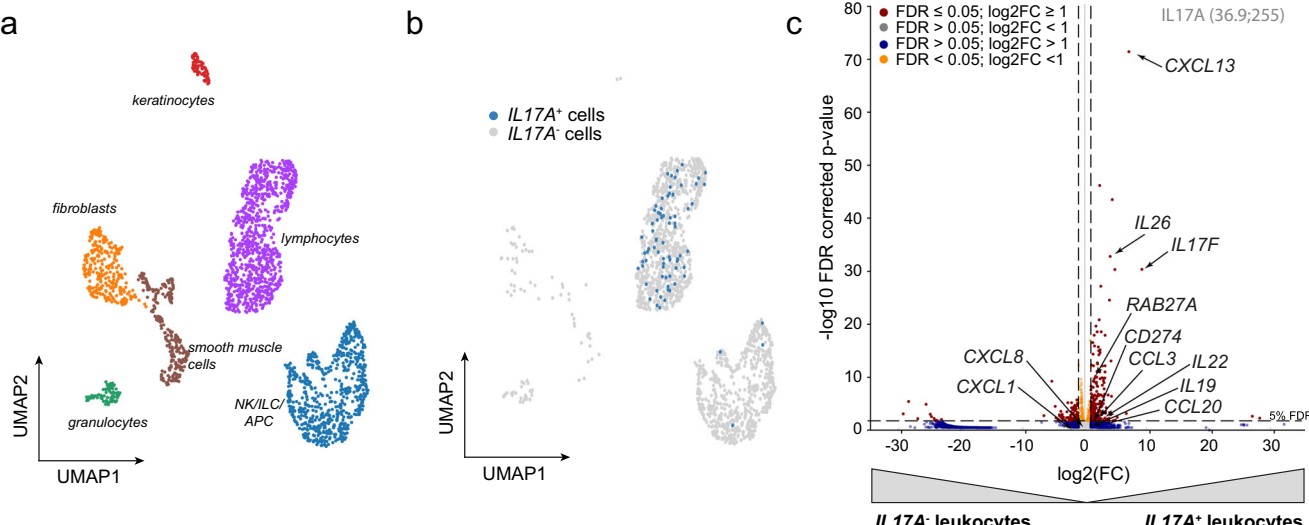

**Fig. 4 | Cytokine transcript-positive single cells express immune cell derived genes, but no tissue markers. a** UMAP (Uniform Manifold Approximation and Projection) plot indicating the composition of cellular clusters in the single-cell (sc) RNASeq dataset derived from psoriasis patients. **b** scRNA-seq analysis of psoriasis skin highlighting *IL17A* expression in lymphocytes (blue) in a UMAP plot. **c** Volcano plot analysing differentially expressed genes (DEG) in *IL17A* positive (*IL17A*⁺) versus *IL17A* negative (*IL17A*⁻) leukocytes in the scRNASeq dataset of psoriasis. Benjamini–Hochberg was used to determine statistical significance.

matched that of antigens previously described in ncISD. In psoriasis, cytokine transcript-positive leukocytes were almost exclusively found throughout the epidermis, where epidermal and melanocytic auto-antigens of psoriasis are expressed, e.g., ADAMTSL5[18], LL37[19], or lipid antigens presented via CD1[12]. By contrast, antigens reported in LP are located at the interface of the basal epidermis and the upper dermis, e.g., DSG[20], and several Hom s proteins[21] as potential antigens of AD are expressed in a similar location potentially leading to leukocyte activation. Our findings are supported by recent publications investigating cytokine mRNA positive cells in inflamed skin highlighting a predominant expression in the epidermis and a co-localization with *CD3* expression[22–24].

To understand the tissue response profile of cytokine transcripts in inflamed skin, we characterized them in a tissue-dependent manner by implementing the tissue annotations as a covariate. In the spatial context, we identified a reliable response signature of type 3 immune responses in the epithelium mediated by *IL17A*, *IL17F*, and *IL26* to induce markers of oxidative stress such as *NOS2*, neutrophil migration such as *CXCL8*, and antimicrobial peptides like *S100A7A* and *DEFB4A*. By contrast, markers of type 1 immunity were chemokines such as *CXCL9*[25], *CXCL10*, and cytotoxic markers. The role for IFN-γ mediated apoptosis and necroptosis in type 1 ncISD is well established[7,8] and is reflected by the expression of *FASL* and *GZMB* in *IFNG*⁺ spots. Type 2 immunity showed the least well-defined epithelial response signature, mostly built of type 2 attracting chemokines such as *CCL17*, *CCL19*, and CCL22. This signature was exclusively mediated by *IL13* as *IL4* transcripts were virtually undetectable in lesional skin even of AD.

The insight that a few cytokine transcripts build the basis of a massive amplification cascade of responder transcripts explains why response genes rather than the signature cytokines themselves are currently suggested as robust biomarkers for diagnostic or theranostic

purposes in ncISD. Examples are a molecular classifier for differential diagnosis of psoriasis and eczema using *NOS2* and *CCL27*[26,27], prediction of the response to anti-IL-17 therapies in psoriasis by IL-19 levels in serum[28], as well as correlation of the severity of psoriasis with DEFB4A[29] or the severity of AD with CCL17/TARC[30].

A reliable identification of disease-driving immune cells and their cognate antigen might pave the way for curative treatment strategies of ncISD, e.g., antigen-specific immunotherapy. This has been attempted e.g., in AD as a global strategy with modest clinical efficacy[31], most likely because there is the need to identify disease endotypes defined by antigen-specificity of disease-driving immune cells, within this heterogeneous disease. The proof-of-principle that curative therapies of ncISD are possible was made in the autoimmune blistering disease pemphigus vulgaris. Here, the causative antigen desmoglein 3 (DSG3) is identical in most patients and it is thus possible to design targeted therapies for the whole patient group. In fact, modified CAR T cell approaches neutralizing exclusively Dsg3-specific cells resulted in impressive and sustained clinical improvements[32,33].

Our density-based clustering method centers clusters around cytokine transcript-positive spots, and consecutively optimises the radius of considered cytokine-specific responder signatures in each tissue slice according to in vitro stimulation of the epidermis. This enabled us to analyse the impact of detected transcripts on their direct surroundings forming local immune microenvironments and calculate a distinct spatial correlation, independent of sample size and heterogeneous number of cytokines. Bulk and single cell sequencing of lesioned skin suggested that few cytokines may drive inflammation, and this is further strengthened by the observed correlation between cytokine transcripts and responder genes in spatial context, thus giving further evidence of functional regulation. The clustering approach can be generalised from epidermis to dermis when adjusting

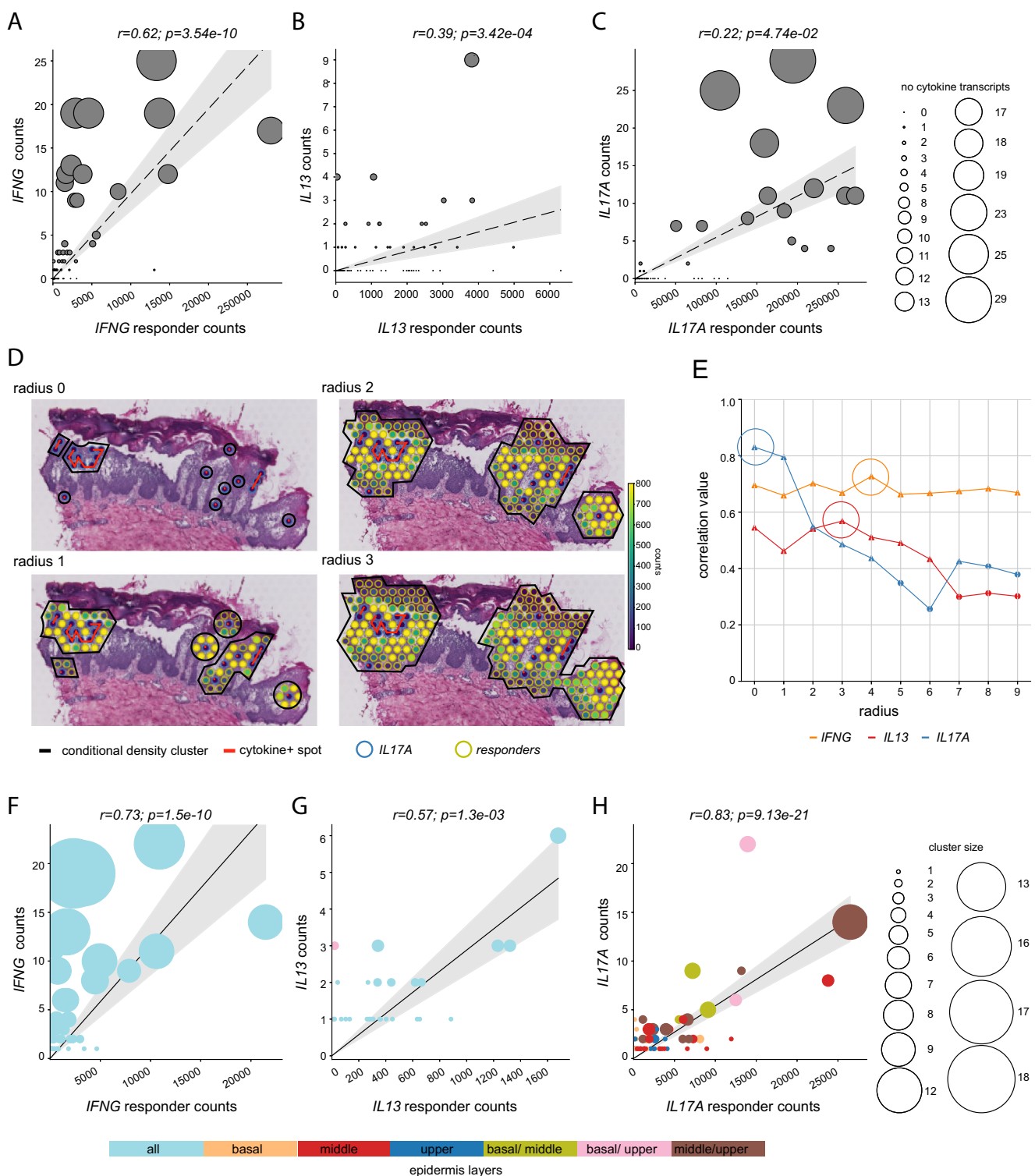

responder signatures, as well as to other diseases and tissues. Our method can be leveraged for identifying biomarkers and disease drivers in the future. Similarly, in-depth evaluation of data-driven cytokine response genes will be a next step to purify distinct response signatures. By integrating three-dimensional spatial information using consecutive tissue sections, the algorithm could be further improved to identify disease-driving networks across tissue sections from the same patient, identifying antigen-specific T cell activation and may highlight promising precise treatment strategies.

A blueprint for successful precision medicine can be found in recent developments in oncology. Typically, tumors such as

malignant melanoma are characterized by thousands of distinct mutations[34]. However, few of them are actually driver mutations leading to tumor growth and metastasis[35]. Targeting these driver mutations by specific targeted small molecules has led to dramatically increased survival rates of melanoma patients in recent years[36]. Here, we demonstrate parallels to inflammatory skin diseases—non-cytokine-secreting immune cells may be seen as irrelevant bystander cells, while targeting cytokine-producing immune cells is a promising strategy for effective and potentially curative treatments of ncISD. A prerequisite is to localize these cells in the inflammatory microenvironment and to identify the specific antigen that disease-

**Fig. 5 | Immune response is spatially correlated with cytokine transcript number. A–C** Weighted Spearman correlation between overall cytokine transcripts and responder transcripts per whole tissue slice in the epidermis. Each point in the plot represents the sum of all cytokine- and responder transcripts in a tissue sample. The size of the points represents the number of observed cytokines on a tissue slice. **D** Representative tissue slice of psoriasis showing *IL17A* expression in relation to its responder signature and different radiuses around the *IL17A*-positive leukocyte spot (Ø55μM). The filling of each circle represents the UMI counts according to the scale on the right capped at 800 UMI-counts for either responder genes (yellow circle) or cytokine transcripts (blue circle for *IL17A*). Red lines connect neighboring *IL17A* transcript-positive spots that together with the surrounding responder gene positive spots create a cluster highlighted by a black line. **E** Weighted Spearman correlation values for *IFNG* (orange), *IL13* (red), and *IL17A*

(blue) depending on the radius from the cytokine transcript-positive leukocyte spot. Strongest correlations for each cytokine are indicated with a circle. Triangles and circles indicate correlations with *p*-values smaller and larger than 0.05, respectively. **F–H** Spatial weighted Spearman correlation incorporating the spatial relation of cytokines and their response located in the epidermis. Shown is the radius for each cytokine with the highest correlation value. This is radius of 4 for *IFNG*, radius of 3 for IL13 and radius of 0 for *IL17A*. Each point in the plots represents the sum of the counts of each cytokine and its responders in a cluster and the size of each point represents the number of cytokine transcripts in a cluster. The color of each spot is associated with the corresponding epidermis layer. Significance in **A–C** and **F–H** was determined by two-sided *p*-value, the line was calculated using the ordinary least square model, the shaded area indicates the 95% confidence interval.

driving immune cells react against, which may pave the way for precision medicine in ncISD.

## Methods

Research performed in this study complies with all relevant ethical regulations. The study was approved by the local ethical committee (Klinikum Rechts der Isar, 44/16 S) and all patients gave written informed consent.

### Study cohort

The study cohort consisted of patients suffering from the non-communicable inflammatory skin diseases (ncISD) psoriasis vulgaris, acute and chronic atopic dermatitis (AD), and lichen planus (LP). Diagnosis was confirmed by a dermatologist on the basis of histological assessment, patient history and clinical phenotype. Included patients did not receive systemic treatment prior to skin sampling. Two independent patient cohorts were used for the study: (1) a bulk- and single cell sequencing cohort consisting of lesional and non-lesional samples of psoriasis (male: $n = 104$, mean age $51,78 \pm 14,24$; female: $n = 84$, mean age $56,04 \pm 18,37$) atopic dermatitis (AD) (male: $n = 63$, mean age $48,83 \pm 17,78$; female: $n = 31$, mean age $48,36 \pm 21,76$), and Lichen planus (LP) (male: $n = 25$, mean age $55,29 \pm 12,31$; female: $n = 33$, mean age $59,05 \pm 14,44$) and (2) spatial transcriptomics cohort including 31 patients. Characteristics of the spatial transcriptomics cohort are given in Supplementary Table 1. Patients were not compensated for study participation.

### Spatial transcriptomics

Tissue sectioning, staining, library preparation: After obtaining non-lesional and lesional skin biopsies (6 mm), one third of each sample was immediately snap frozen in liquid nitrogen. Samples were then stored at −80 °C until cryosectioning. Upon cryosectioning, samples were equilibrated to cryostat (NX70, Thermo Fisher Scientific) chamber temperature for at least 30 min and covered in optimal cutting temperature compound (OCT). Sections were taken at 10 μm thickness at −17 °C and directly placed onto the Visium Spatial Gene Expression slide (10x Genomics). Slides were processed using the Visium Spatial Gene Expression Kit (#PN-1000184, 10x Genomics) following the CG000239 Visium Spatial Gene Expression Reagent Kits—User Guide RevA. Optimal experiment conditions were investigated using the Visium Spatial Tissue Optimization Kit (#PN-1000193, 10x Genomics) on independent healthy, lesional and non-lesional skin samples, following the CG000238 Visium Spatial Gene Expression Reagent Kits—Tissue Optimization Rev A. To perform HE staining, samples were incubated in Mayer's Hematoxylin (Dako) for 2 min and Eosin (Sigma) for 40 s, while Bluing buffer was omitted. Sections were permeabilized for 14 min and imaged using the Metafer Slide Scanning Platform (Metasystems) or the IX73 Inverted Microscope Platform (Olympus). Raw images were processed using VSlide 4.3 software (Metasystems). Libraries of the individual datasets were pooled together separately and thereafter sequenced by the National Genomics Infrastructure

(NGI, Sweden) on the Illumina NovaSeq platform using the recommended 28-10-10-120 cycle read setup.

Sample annotation: HE images of corresponding samples were evaluated and annotated manually by two trained dermato-pathologists in a blinded manner using Loupe Browser (10x Genomics). Spots being present on tissue parts that were clearly destructed and broken off the section were marked and excluded from any further analysis. Samples were annotated for general morphology, anatomical structures, and specific cell types. Regarding general morphology, spots were categorized as "epidermis" or "dermis". Spots that were localised at the dermo-epidermal junction were additionally marked as "junction". Epidermal spots were moreover classified as "upper epidermis", "middle epidermis" or "basal epidermis". To make the position of spots within the dermis comparable across the whole dataset, all spots categorized as "dermis" were further divided into "dermis 1" to "dermis 7" indicating the depth of the dermal layer in a standardized fashion.

Data processing: 62,968 spots were sequenced and samples were processed using 10x Visium Space Ranger-1.0.0. Quality control (QC) measures were applied on 90 samples with 82 passing QC. The sections were normalised and batch correction was applied to account for variances between the slides. DEG and pathway enrichment analysis were performed. Finally, the correlation between cytokine-secreting leukocytes and cytokine-dependent responder genes was investigated via a pseudo-bulk aggregation and a spatially weighted correlation approach.

Due to acute inflammation, a high mitochondrial fraction was anticipated, thus a conservative 25% cut-off was chosen. Spots with a minimum of 30 detected genes, and genes which were observed in at least 20 spots were considered. In addition, the QC enforced a minimum and maximum UMI-count of 50 and 500,000, respectively. The data were normalised using size factors calculated using the 'scran' R-package[37], log10 transformed, and a pseudo count of one was added to avoid log-transformation of zero[38]. Highly variable genes were selected batch independently using 'SCANPY-1.9.1's' highly_variable_gene function with flavor cellranger[39]. The ST dataset was batch corrected with 'scanorama'[40] accounting for the variances between the projects. Further, the dataset was dimensionally reduced by applying a principal component (PC) analysis with n_pcs = 15 and embedded in a neighborhood graph with n_neighbors = 15. Subsequently, the data were represented in a 2D UMAP plot.

The mean number of spots per section was $767 \pm 293$. Here, lesional skin was represented by significantly more spots per section ($823 \pm 324$ vs $633 \pm 125$, $p = 0.0015$) which reflects morphological changes in the tissue due to inflammation. This was further supported by the UMI counts per spot per section being $3189 \pm 6620$ in lesional skin and $605 \pm 613$ in non-lesional skin ($p = 0.0002$) (Supplementary Table 2).

Clustering of transcriptomes: The ST analysis benefited from expert annotations of dermato-pathologists, thus forming the clusters based on epidermis layers, junction and dermis depths 1-7. For the

clustering of the scRNA-seq data, we leveraged the Leiden algorithm and determined the number of clusters by the maximum silhouette score, and prior knowledge, i.e. enriched marker genes in stable clusters. At a resolution of 0.1, the maximum silhouette score was 0.54.

Spatial enrichment of cytokines in specific skin layers: Unnormalised count matrices, and a targeted analysis scrutinised for *IL17A*, *IFNG*, and *IL13* was used to analyse cytokine expression compared to a housekeeping gene, *GAPDH*, in ST. Cytokine expression levels were quantified within the manually curated skin layers, and significant spatial enrichments were tested with Wilcoxon signed-rank test.

Differential gene expression (DEG) and pathway enrichment analysis: To characterize cytokine expressing cells and spots, leukocytes were defined by expression of least one of the marker genes *CD2*, *CD3D*, *CD3E*, *CD3G*, *CD247 (CD3Z)*, and *PTPRC (CD45)* or combinations of these markers in the ST and single-cell datasets. Presence of one transcript was regarded as a positive association. Leukocytes were defined as cytokine-positive if at least one UMI-count of the cytokine gene was detected. Prior to the DEG analysis, the counts were normalised using size factors calculated on the whole dataset.

Genes characterising cytokine-positive spots were compared with cytokine-negative spots to obtain differentially expressed genes (DEG) on a spot-level using 'glmGamPoi'[41] and the multiple testing method 'Benjamini–Hochberg' (BH). In addition to the unnormalised counts, the calculated size factors were provided and biological variances were included as fixed effects in the design matrix. In the design matrix the covariates cellular detection rate (cdr), patient, and annotation were included. This enabled to account for variances between the fraction of genes being transcribed in a cell[42], and the difference in gene expression between cells that are located in different tissue types and are of different cell types, respectively. The following model was used for the ST dataset

$$Y_{sg} \sim cdr + project + patient + annotation + condition \qquad (1)$$

or for the single psoriasis patient scRNA-seq dataset

$$Y_{sg} \sim cdr + annotation + condition, \qquad (2)$$

where $Y_{sg}$ is the raw count of gene $g$ in the cell or spot $s$. A gene is called significantly differentially expressed if it meets the cut-off parameters of $p$-value $< = 0.05$ and $|log2FC| > = 1$.

Pathway enrichment analysis was performed using the Bioconductor 3.16 packages 'ReactomePA'[43] and 'org.Hs.eg.db'[44] and illustrated using the Bioconductor 3.16 package 'enrichplot'[45]. The $p$-values of the pathways were corrected using the BH method and a $p$-value and $q$-value cut-off of 0.05 was applied.

Experimentally derived cytokine responder gene signatures: Responder gene signatures to type 1, type 2, and type 3 mediated inflammation were developed by stimulating primary human keratinocytes in vitro with recombinant IFN-γ, IL-13 and IL-17A (20 ng/ml each), respectively (Supplementary Fig. 7A). After 16 h, total RNA was isolated and whole genome expression arrays (SurePrint G3 Human GE 8X60K BeadChip (#G4858A-028004, Agilent Technologies)) were performed according to the manufacturer's instructions. Gene expression data was filtered for $p$-value $< 0.05$, adjusted $p$-value $< 0.05$, and log2FC $> 1.5$ for IL-17A and IFN-γ or log2FC $> 1$ for IL-13. To further identify the most relevant responder genes in vivo, gene expression of cytokine transcript-positive spots residing in the epidermis was compared with the respective differentially expressed genes of in vitro stimulated keratinocytes (e.g. DEG of *IL17A*+ spots with IL-17A stimulated keratinocytes). The overlapping gene signature was then curated for genes being present in the response signature of all cytokines (e.g. IL-17A-specific genes in the IFN-γ response signature and vice versa). This, however, did only lead to exclusion of four IL-13-specific genes that were also present in the IFN-γ response signature. Hereby, 21, 29

and 4 responder genes could be identified for IL-17A, IFN-γ and IL-13, respectively (Supplementary Fig. 7B–D). To not rely on only 4 genes, well-known, literature-based genes were added to the IL-13 response signature (Supplementary Fig. 7).

Spatial correlation: To evaluate the correlation between cytokine transcripts and responder signatures induced in surrounding tissue cells, we annotated ST spots as either cytokine transcript-positive (containing at least one cytokine count), or alternatively as other. As the responder gene signature was obtained from in vitro stimulated primary human keratinocyte experiments, the correlation analysis focused solely on the epidermis. For leveraging the full power of ST, we data-driven defined close proximity to cytokine transcript-positive spots based on a density based clustering method (Fig. 5D, Methods).

Density-based clustering: We developed a density-based clustering method that leverages as seeds confirmed cytokine transcript-positive spots. For this, we anchored on cytokine transcript-positive spots, and expanded clusters by spots in their neighbourhood. The neighbourhood of a cytokine transcript-positive spot is defined by a radius, which is equal or larger than zero. In addition, clusters were merged if they overlapped. The radius was individually optimised for each cytokine, i.e. we explored different radiuses for *IFNG, IL13* and *IL17A* in a range from 0 to 9. The optimal radius is chosen by maximising the correlation between cytokine-positive clusters and their corresponding responder gene signatures. By applying these conditions, the clusters were characterized based on the density of cytokine-positive spots and a fitted radius. In more detail, the adjacent cytokine transcript-positive locations were obtained per sample using the KDTree algorithm[46] with the Euclidean metric and a maximum distance of 2.0. For this purpose, the index array provided by 10X Genomics was used. Afterwards, the locations of adjacent cytokine-positive mRNA capturing points were connected using a graph as backbone. Here, the nodes were the cytokine-positive spots and the edges equal the distance between the spots. Moreover, the nearest neighbour spots were determined by

$$C_n = \sum_{j=-r}^{r} \sum_{i=-2r+|j|}^{2r-|j|} s_{ji} \qquad (3)$$

where $r$ is the radius of the cluster and $s_{ji}$ is the nearest neighbour spot in row $j$ and column $i$. Then the cytokine-positive graph was merged together with the nearest neighbour responder spots resulting in an agglomerated graph. Finally, the counts of responder genes and cytokines in each cluster were read out and a weighted Spearman correlation was calculated. The weights were determined by the measured transcripts in the graph to account for the size and impact of the density cluster. A line was fitted between cytokine counts and responder genes using an intercept of 0.

Identification of cytokine-related genes: For identifying cytokine-related genes within ST, we leveraged our density-based clustering method. We used the normed and batch corrected epidermal ST data and performed a DEG analysis between cytokine transcript-positive spots and spots which were not included in the optimal radius density clusters, 4, 3, and 0 for IFNG, IL13, and IL17A, respectively. 'glmGamPoi'[41] and the design matrix

$$Y_{sg} \sim cdr + project + patient + annotation + condition, \qquad (4)$$

where s defines a spot and g is the raw count of a gene, were used to perform the DEG analysis. For details on variable definition, see section on DEGs above. Accordingly, $p$-values were corrected using BH. DEGs were determined requiring cut-offs of $|log2FC| > 1$ and FDR corrected $p$-value $< 0.05$.

Pathway enrichment analysis was performed using the Bioconductor 3.16 packages 'ReactomePA'[43] and 'org.Hs.eg.db'[44].

Illustration of the enriched pathways are created using the Bioconductor 3.16 package 'enrichplot'[45]. The p-values of the pathways were corrected using the BH method and a p-value and q-value cut-off of 0.05 was applied.

Spot deconvolution: We used Tangram[17] to identify cell types in a spot leveraging as reference the preprocessed public scRNA-seq dataset[47]. Tangram[17] spatially aligns the single cell expression of the cell types matching it to the expression profile in the spots.

Lesion and non-lesion samples from Pso and AD were extracted and provided, together with a specimen from the ST data, as input to the Tangram algorithm. The model was trained using the mode 'clusters' (tissue_layer), cluster_label 'subclass_label' (full_clustering) and a 'rna count based density prior'. The number of epochs was set to 400 and a CPU was used.

## In situ hybridization

In situ hybridization was performed using the RNAScope Multiplex Fluorescent V2 Assay for paraffin embedded tissue sections (#323135, Advanced Cell Diagnostics, Newark, CA) on lesional skin sections of psoriasis, AD, and LP (5 μm each). The assay was performed using probes designed by ACD targeting human *IL17A* (#310931-C2), *IFNG* (#310501) or *IL13* (#586241-C3) mRNA. Positive control sections were prepared using human peptidylprolyl isomerase B (PPIB) probe whereas negative controls were assessed using bacterial gene probes. Briefly, target probes were hybridized followed by signal amplification according to manufacturer's protocol. Each probe was stained by Opal 690 (Akoya Biosciences, Marlborough, MA) using a single-plex setup. Subsequently, skin sections were examined using microscope slide scanner (Axio Scan.Z1 Zeiss, Germany) at 20x magnification. Then, images were visualized using QuPath-0.3.2 software[48]. Images were individually evaluated by two trained dermato-pathologists in a blinded manner. Cells were counted positive if punctate-dot RNAscope signal co-localized with nuclear staining.

## Immunohistochemistry

5 μm sections of paraffin embedded skin samples were air-dried overnight at 37 °C, dewaxed and rehydrated. Stainings were performed by an automated BOND system (Leica) according to the manufacturer's instructions: epitope retrieval was performed at pH6 in epitope retrieval solution (DAKO) and incubated with goat anti-human IL-17A (#AF-317-NA, R&D Systems) followed by a biotinylated anti-goat secondary antibody (#BA-9500-1.5, Vector Laboratories). For detection of specific binding, streptavidin peroxidase and its substrate 3-amino-9-ethyl-carbazole (DAKO) were used. All slides were counter stained with hematoxylin. Stainings without primary antibodies were used as negative control. Positive cells were counted in four to nine visual fields per condition.

## Isolation of primary human T cells and in vitro stimulation

Peripheral blood mononuclear cells were isolated from peripheral blood of healthy donors (n = 1 male, n = 2 female, age 38 ± 7) by density centrifugation. Primary human Pan T cells were then isolated using magnetic beads (Pan T cell isolation kit, #130-096-535, Miltenyi Biotec), followed by CD4 (human CD4 microbeads, #130-045-101, Miltenyi Biotech) or CD8 (human CD8 microbeads, #130-045-201, Miltenyi Biotec) isolation. Defined numbers of cells were stimulated with platebound anti-CD3 and anti-CD28 antibodies (0.75 μg/ml; # 555329, # 555329, BD Biosciences) for 10 min, 1 h or 6 h, or were left unstimulated. Stimulated T cells were collected after 10 min, 30 min, 1 h, 6 h, 12 h, or 24 h stimulation and RNA was isolated for subsequent real time PCR analysis with the following primers: *IL17A* (fw: CAATCCCCAGTTGATTGGAA; rev: CTCAGCAGCAGTAGCAGTGACA), *IFNG* (fw: TCAGCCATCACTTGGATGAG; rev: CGAGATGACTTCGAAAAGCTG), *IL13* (fw: TGACAGCTGGCATGTACTGTG; rev: GGGTCTTCTCGATGGCACTG),

18 S (fw: GTAACCCGTTGAACCCCATT; rev: CCATCCAATCGGTAGTAGCG).

## Flow cytometry of skin T cells

Primary human T cells (n = 52) were isolated by digestion of fresh human skin biopsies (Ø 6 mm) in RPMI containing FCS, Collagenase type IV (Worthington), and Deoxyribonuclease I (Sigma) at 37 °C overnight followed by dissociation using the gentleMACS Dissociator (Miltenyi Biotec). Freshly isolated skin T cells were passed over a cell strainer and directly used for flow cytometric analysis. For flow cytometric analysis, T cells were stimulated with PMA/Ionomycin (10 ng/ml and 1 μg/ml, respectively) (both Sigma) for 5 h in the presence of Brefeldin A and Monensin (both BD Biosciences). Surface staining was performed at 4 °C and followed by fixation/ permeabilization using the fixation/permeabilization kit (BD Biosciences). Staining of intracellular cytokines was performed at room temperature. Antibodies used were CD3-Bv650 (#563852, clone UCHT1, dilution 1:50), CD4-BV421 (#562842, clone L200, dilution 1:20), CD8-APCCy7 (#557834, clone SK1, dilution 1:20) (BD Biosciences), IL-17A-PeCy7(#512315, clone BL468, dilution 1:20), IFN-γ-PerCPCy5.5 (#506528, clone B27, dilution 1:100), TNF-α-BV510 (#502950, clone MAB11, dilution 1:100) (BioLegend), IL-22-Pe (#12-7229-41, clone 22URTI, dilution 1:20, eBioscience), IL-10-APC (#130-108-135, clone JES3-9D, dilution 1:10, Miltenyi Biotec). Flow cytometry data was analysed and visualized using FlowJo 10.7.1.

## Single-cell RNA sequencing

A lesional skin sample (6 mm) was taken from a psoriasis patient and digested immediately for 3 h at 37 °C using the MACS whole skin tissue dissociation kit (Miltenyi Biotec) and the gentleMACS Dissociator (Miltenyi Biotec) according to manufacturer's protocol. The obtained cells were stained for CD3 (#300450, Biolegend,) and CD45 (#563880, BD Biosciences) and sorted using a FACSAria Fusion (BD Biosciences). Here, dead cells and doublets were gated out and cells sorted based on size (FSC/SSC) and CD3/ CD45 expression into three populations: skin cells (keratinocytes), T cells (CD45+, CD3+), and APCs (CD45+, CD3-). The obtained cells were mixed in equal ratio (1:1:1) to a final cell number of 16,000 and used as input for the sc library generation by the 10x Genomics kit (Chromium Next GEM SingleCell 3′ GEM, Library & Gel Bead Kit v3.1, #1000121) according to the manufacturer's protocol. The libraries were sequenced on an Illumina HiSeq4000 via paired-ends with a read length of 2 × 150 bp at a sequencing depth of 40 million reads.

scRNA-seq data processing: The pre-processing and QC of our scRNA-seq data was identical to ST, besides enforcing a minimum of 500 genes per cell, and a minimum and maximum UMI-count of 600 and 25,000, respectively. In addition, according to the scrublet pipeline[22], no doublets were detected. Additionally, no batch effect was detected in the scRNA-seq data and the number of PCs was set to n_pcs = 7. On the public dataset from Reynolds, Gary, et al. "Developmental cell programs are co-opted in inflammatory skin disease".[47] we required a minimum of 250 genes per cell to be measured with at least 1 UMI-count. Further, a cell should have minimum and maximum UMI-count of 500 and 400,000, respectively. The ribosomal fraction should be in the range of 5% to 60% and the maximum MT-fraction was set to 25%. Doublets were removed using scrublet having a score above 0.6. The data was normalized using SCANPY-1.9.1 and 4000 HVG were determined per sample. Further, PCA was applied with n_pcs = 10 and the dimensionally reduced data was embedded in a neighborhood graph with n_neighbors = 15. Subsequently, the data were represented in a 2D UMAP plot.

## Reporting summary

Further information on research design is available in the Nature Portfolio Reporting Summary linked to this article.

## Data availability

RNA sequencing data can be obtained at GEO (www.ncbi.nlm.nih.gov/geo accession number: GSE206391). This study did not generate new unique reagents or use publicly available datasets. Source data are provided with this paper.

## Code availability

Source code is available at github: https://github.com/Chillig/ST_biostatistical_analysis, and zenodo [https://zenodo.org/record/7309851#.Y3HXGy1Q1QI][49].

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

## Acknowledgements

The authors acknowledge support from the National Genomics Infrastructure in Stockholm funded by Science for Life Laboratory, the Knut and Alice Wallenberg Foundation and the Swedish Research Council, and SNIC/Uppsala Multidisciplinary Center for Advanced Computational Science for assistance with massively parallel sequencing and access to the UPPMAX computational infrastructure. The authors thank Thomas Walzthoeni for processing the sequencing data, and Carlos Talavera López, Malte Lücken, Elmar Spiegel, Ronan Le Gleut, and Giovanni Palla for discussions and valuable feedback. Furthermore, we thank Life Science Editors for revising the manuscript. This work is supported by Deutsche Forschungsgemeinschaft (DFG) through TUM International Graduate School of Science and Engineering (IGSSE) (CH, M.P.M, M.M., S.E.).

## Author contributions

Conceptualization: A.S., C.H., K.E., M.M., S.E.; methodology: A.S., C.H., A.F., N.B., J.T., M.M., M.J., N.K., A.C.P., E.S.; N.G.-S.; visualization: C.H., S.E., S.F., A.S.; funding acquisition and supervision: K.E., M.P.M., S.E.; project administration: A.S., C.H., A.F.; writing draft: K.E., A.S., C.H., M.P.M., S.E., A.F.; review and editing: M.S., C.B.S.-W., T.B., F.T.

## Funding

## Competing interests

The authors declare no competing interests.
