## [Peer Review File · Nature Communications]

Spatial transcriptomics landscape of non-communicable inflammatory skin diseasesREVIEWER COMMENTS

Reviewer #1 (Remarks to the Author):

The top level goal of this manuscript was application of a recently developed spatial transcriptomics platform to understand inflammatory transcript expression in human skin inflammatory disease. The authors selected and analyzed 5 psoriasis, 5 atopic dermatitis, 5 lichen planus, and 3 pityriasis rubra pilaris samples.

Across these 18 patients, the analyzed histopathology samples included 64 samples, which consisted of 18 lesions samples in duplicate (36 samples) and 14 non-lesional controls, which appeared from Methods to be from the same patients (28 samples). Collectively, spatial transcriptomes of about 52K spots were retrieved, for which after filtering, about 28K lesional and 15K non-lesional spot-transcriptomes were further analyzed.

The authors describe two means of analyzing the consequent data. In "Workflow 1", they state that they incorporate "spatial features in differential gene expression (DEG) analysis of spots containing cytokine transcript positive versus cytokine transcript-negative leukocytes, followed by pathway enrichment analyses". In a "Workflow 2", they label "cytokine transcript-positive spots and then use a density-based clustering methods to boost correlations of cytokine and responder gene signatures according to spatial features."

Their conclusion is that single copies of cytokine transcripts amplify >1000 "responder transcripts" which are causative for disease in the tissue microenvironment. They then attempt to validate this hypothesis using a variety of patient cohorts and techniques such as in situ hybridisation, single-cell and bulk sequencing, immunohistochemistry, flow cytometry and cell culture analysis.

I had an exceedingly difficult experience trying to follow the reasoning and methodology of this study. The data is presented very elliptically, as if the methodology is well-established, when this is not the case. As one of many examples, but one that is illuminating, the idea of a 'spot' is presented and then used without elevating or discussing the obviously important issue of how many cells are captured in a spot and how this resolution relates to a complex like human epidermis where that spot might encompass APCs, T cells, keratinocytes, neural cells, etc.

I interpret "Workflow 1" to mean that that transcriptomes of spots in which both T cell markers and (leukocytes???) cytokine expression was detected were compared to spots with T cell markers in which cytokines were not detected, and pathway analysis was performed on the comparison. I interpret "Workflow 2" to mean that in spots (with or without T cell markers?) positive for cytokines, analytic methods were used to amplify and/or assess correlation between cytokine expression and so-called responder genes.

Among the questions I could not answer from the manuscript or even a reading of the Supplemental Methods. These need to be front and center in the actual manuscript.

- How were 'responder' genes defined? Is it the keratinocyte produced transcripts on provocation in lines 188-190 (table s6), well after extensive discussion of responder genes in the Results analysis? Or a different manual curation by the authors or an external method as implied in line 156? Are the genes responsive in terms of being directly induced genes, or simply also enriched in the Th subtype?
- What are the full list of T cell marker genes used to call a spot positive for leukocytes? Were T cell subtypes considered?
- Do the authors believe a given T cell positive spot contained multiple T cells or just one? Does it matter? How would it affect the analysis if different types of T cells were found at a specific address?

There is a major emphasis in the writing of this manuscript on there being a "single transcript" in a spot or T cell to the point that this is stated in the abstract. I am not sure this is even necessary to the

most interesting point of the paper (numerous downstream target genes clustering spatially near Th cytokine detect). I found this assertion difficult to validate from the presented data and the approaches suspect, for example "Single-cell RNASeq analysis of psoriasis also indicated few transcripts per IFNG+ or IL-17A+ cell" on lines 125/126- are the authors using a notoriously shallow and low-sensitivity method (scRNA-seq) to validate their claim that detection of single transcripts is representative of the actual per cell expression?

Immune response is spatially correlated with cytokine transcript number

If I understand this section, the authors are able to show that DEGs directly resulting from cytokine shifts (as per keratinocyte in vitro experiments) are found in spatial relation to addresses where that cytokine is detected. This is interesting and would have served more logically as the focal point of the paper. What is the resolution of these spatial hotspots compared with the hotspot of the technology? Are the differences distinct between diseases?

Minor points

- It is really not optimal to apply a new large scale profiling approach like this on so few samples of each disease class. A single disease class would have been informative and the data much more interpretable.
- Relatively limited information on clinical samples – PASI/EASI severity, lack of recent treatment, formal histopathology
- Line 108: "the tissue inflammation of ncISD is driven by T cell cytokines" – I believe that many researchers believe APCs and keratinocytes play a major role here.

Reviewer #2 (Remarks to the Author):

The manuscript by Schabitz and Hillig et al. characterize the spatial transcriptomic landscape of lichen planus, atopic dermatitis, psoriasis, and pityriasis rubra pilaris. They find general low cytokine expression from a few T cells that initiate a larger response in neighboring cells, which is validated using other sequencing methods, in situ hybridization, IHC, and flow cytometry. This is an interesting report that suggests active skin disease is maintained by a few active T cells. This work, along with the abundance of sequencing data, will be a valuable resource to the skin field once the data is publicly available. A few clarifications are needed to help the reader better understand the results as detailed below.

1) The sequencing data needs to be deposited in a publicly available database. The authors have boilerplate language that suggests this will be the case, but no specific accession number is provided.

2) Going through the figures to understand the underlying data is a bit difficult given some of the necessary data is in supplementary figures. I suggest adding Figure S2A into the beginning of Figure 3 and Figure S4A into the bottom part of Figure 3. In addition, Figure 4D-F show representative cytokine expressers and their responder signatures in tissue slices. It would be helpful to denote where T cells are located to see if all T cells are active or only a subset within the tissue. Also, can the boundary between the epidermis and dermis be denoted by a line? It is difficult to determine where the boundary is with the superimposed spots.

3) This figure legends could use more detailed information to describe what is shown in the figure panels. For instance, there are white and red circles in Figure 2F. The red circle is likely the blown up image, but I'm not sure what the white circles represent. There are also red lines connecting cells in Figure 4D and F and an orange line on one end of the conditional density cluster that remain a mystery.

4) Do the cytokine+ spots in the dermis also show positive correlation with responders?

Reviewer #3 (Remarks to the Author):

The authors have conducted an impressive analysis of spatial sequencing data from a group of inflammatory skin diseases. They found that a relatively small number of T cell cytokine producing cells are present in the epidermis and upper layers of the dermis that distinguish different diseases. Importantly, the authors have developed a novel bioinformatics approach to identify cytokine responder genes in the adjacent spatial landscape.

1. Impact of the manuscript.

a. The author state in the abstract that "Despite the expected T cell infiltration, we observed a rather low frequency of only 1-10 pathogenic T cell cytokine transcripts per skin section." This appears to have been reported previously using in situ hybridization (see Comment #4) in psoriasis and atopic dermatitis.

b. The others further state: "Nevertheless, cytokine expression was limited to lesional skin, presented in a disease-specific pattern and evoked specific responder signatures in direct proximity of cytokines. This showed that single cytokine transcripts initiate amplification cascades of thousands of specific responder transcripts forming localized epidermal clusters. Thus, within the abundant and heterogeneous T cell infiltrates of ncISD, only a few T cells drive disease by initiating an inflammatory amplification cascade in their local microenvironment." It is certainly known that T cell cytokines act in trans- for example IFN-gamma and IL-17 have downstream effects on keratinocytes. However, the authors developed a novel approach to study the downstream effects of T cell cytokines in adjacent cells by density-based clustering. This methodological advance is significant and should be emphasized. The authors leverage this approach to demonstrate that T cell cytokine downstream responder genes are enriched in adjacent spots. Is it possible to use this approach to identify novel genes associated with the T cell cytokines that are possible downstream responder targets?

2. UMIs vs. spots. The authors report "Despite the expected T cell infiltration, we observed a rather low frequency of only 1-10 pathogenic T cell cytokine transcripts per skin section." They further state "Taking the whole section into account, we detected only a few transcripts for IFNG, IL13 or IL17A (272, 57, or 92 UMI counts in all sections, respectively) in lesional skin (Fig. 2E)." There were 18 lesional samples, but we don't know the distribution of the UMI counts for each sample as well as how many spots were detected with the distribution of UMIs per spot. Should these data be normalised for the size of the biopsy as they are counting the entire section which may vary, or perhaps they were of uniform size? What is the variability of the data? Are the majority of UMIs coming from 1-2 samples? We can infer the minimum number of cells by the number of spots. If there is one UMI in a spot, then there is one cell, if more UMIs per spot, there could be more cells but there is at least 1.

3. Validation by in situ hybridization. The authors state for their in situ hybridization validation that "The median number of transcript-positive cells per section for IFNG, IL13, and IL-17A mRNA were 16, 2, and 0 for LP, AD, and psoriasis, respectively, thus confirming our observations from the ST analysis (Fig. 2G)." They also provide similar data for mean UMI counts per biopsy for scRNA-seq data. Should this be normalised to the size of the section? But this cannot be directly compared to the spatial sequencing data. Also, why is the mean for IL 17A "0" in psoriasis? Isn't this a Th17 disease? Its not clear what the above numbers exactly refer to.

4. In situ hybridization as compared to the literature. I easily found two relevant articles that were not quoted (see below), and it would seem that they find more cytokine+ cells, albeit it is hard to

compare as the calculations are different. Is it possible that there is dropout in the authors' data?

a. JID Innovations, Volume 1, Issue 2, June 2021, 100021, Alice Wang et al, JID Innovations, Cytokine RNA In Situ Hybridization Permits Individualized Molecular Phenotyping in Biopsies of Psoriasis and Atopic Dermatitis

b. British Journal of Dermatology, Volume: 185, Pages: 585-594. Signalling of multiple interleukin (IL)-17 family cytokines via IL-17 receptor A drives psoriasis-related inflammatory pathways. M.A.X. Tollenaere, J.

5. Responder genes-specificity. In Fig 3B for example, the authors show that IL17A responder genes are enriched in IL17A+ spots, but many genes are unlabeled. The specificity of this response would be made clear by showing that IFN-gamma and IL-13 responder genes were not enriched. Also, if the detection of cytokine and responder genes is robust, shouldn't the key cytokine receptor genes be detected as well? The genes encoded receptors for IFN-gamma, IL-13, IL17A should be detected in Fig. 4D, E, F. Also, there is no information on the cell types found in cytokine+ spots or the adjacent responder spots, since each spot will be a mixture of cells.

6. Validation by scRNA-seq. "Confirming the spatial dataset, most leukocyte-associated genes were also identified in the single-cell data, whereas the responder gene signatures were widely missing in the single-cell data (Fig. 3F, Fig. S4C, D)." It is not clear which responder genes are missing in the single-cell data to assess how widely they are missing. Can imputation be used to demonstrate this? Also, the authors previously stated "T cell genes associated with IL17A were IL17F, IL22, and IL26, and the responder signature of IL17A consisted of e.g., IL19, NOS2, S100A7A, DEFB4A, CXCL8, and IL36G (Fig. 3B, C)", but are showing different genes in Fig. 3F.

7. Leukocytes. "To characterize cytokine expressing cells, leukocytes were defined by the marker genes CD2, CD3D, CD3E, CD3G, CD247, and PTPRC in the ST and single-cell datasets. Leukocytes were defined as cytokine-positive if at least one UMI-count of the cytokine gene was detected." Aren't these T cells?

8. The authors state "To characterize cytokine expressing cells, leukocytes were defined by the marker genes CD2, CD3D, CD3E, CD3G, CD247, and PTPRC in the ST and single-cell datasets." It is not clear what quantitative criteria was used for this definition.

Point-to-point reply

Reviewer #1 (Remarks to the Author):

The top level goal of this manuscript was application of a recently developed spatial transcriptomics platform to understand inflammatory transcript expression in human skin inflammatory disease. The authors selected and analyzed 5 psoriasis, 5 atopic dermatitis, 5 lichen planus, and 3 pityriasis rubra pilaris samples.

Across these 18 patients, the analyzed histopathology samples included 64 samples, which consisted of 18 lesions samples in duplicate (36 samples) and 14 non-lesional controls, which appeared from Methods to be from the same patients (28 samples). Collectively, spatial transcriptomes of about 52K spots were retrieved, for which after filtering, about 28K lesional and 15K non-lesional spot-transcriptomes were further analyzed.

The authors describe two means of analyzing the consequent data. In “Workflow 1”, they state that they incorporate “spatial features in differential gene expression (DEG) analysis of spots containing cytokine transcript positive versus cytokine transcript-negative leukocytes, followed by pathway enrichment analyses”. In a “Workflow 2”, they label “cytokine transcript-positive spots and then use a density-based clustering methods to boost correlations of cytokine and responder gene signatures according to spatial features.”

Their conclusion is that single copies of cytokine transcripts amplify >1000 “responder transcripts” which are causative for disease in the tissue microenvironment. They then attempt to validate this hypothesis using a variety of patient cohorts and techniques such as in situ hybridisation, single-cell and bulk sequencing, immunohistochemistry, flow cytometry and cell culture analysis.

I had an exceedingly difficult experience trying to follow the reasoning and methodology of this study. The data is presented very elliptically, as if the methodology is well-established, when this is not the case. As one of many examples, but one that is illuminating, the idea of a ‘spot’ is presented and then used without elevating or discussing the obviously important issue of how many cells are captured in a spot and how this resolution relates to a complex like human epidermis where that spot might encompass APCs, T cells, keratinocytes, neural cells, etc.

We thank the reviewer for this constructive criticism and implemented it in the revised manuscript by adding additional information on the technique ‘spatial transcriptomics’, the spot composition and newly developed bioinformatic tools in the results section (“Gene expression was measured in H&E stained skin sections in so-called spots that are distributed equally over the whole tissue section. The diameter of each spot-diameter is 55 μM, and every spot is distanced to the neighboring spot center-to-center by 100 μM. Given the spot size, a single spot can contain several cells such as immune and epithelial cells. The composition of cell types within a spot can be predicted with deconvolution algorithms, e.g. Tangram (Biancalani et al. 2021)”). In addition, we added this

limitation of spatial transcriptomics using spots with a diameter of 55µm as outlook to the discussion (“Further refinement needs to be implemented in terms of sensitivity of sequencing methods and the spatial resolution that with a spot diameter of 55µm captures more than just one cell. So far it cannot be ruled out that presence of one cell type does affect the activity and response of another cell type. Deconvolution of spatial spots using algorithms like Tangram (47) helps to understand the cellular composition of spots, however, a reduction of the spot diameter is highly desirable and may well be the next evolutionary step in spatial transcriptomics.”). *In particular, we investigated the cellular spot composition of cytokine positive spots in Fig S3 using the Tangram algorithm (see Figure below; Biancalani T, Nat Methods. 2021 Nov;18(11):1352-1362. doi: 10.1038/s41592-021-01264-7. Epub 2021 Oct 28. PMID: 34711971).*

I interpret “Workflow 1” to mean that that transcriptomes of spots in which both T cell markers and (leukocytes???) cytokine expression was detected were compared to spots with T cell markers in which cytokines were not detected, and pathway analysis was performed on the comparison. I interpret “Workflow 2” to mean that in spots (with or without T cell markers?) positive for cytokines, analytic methods were used to amplify and/or assess correlation between cytokine expression and so-called responder genes.

Yes, the reviewer is absolutely correct with his/her interpretation of the two workflows. We apologize for the blurry use of “T-cell” and “leukocytes”, which we believe contributed to confusion. To enhance the accuracy of our description, we now clearly specify that leukocytes were included into the analysis acknowledging not only cytokine producing T cells, but also e.g. innate lymphoid cells and NK cells. The leukocyte spots were defined as being positive for CD2, CD3D, CD3E, CD3G, CD247 (CD3Z), and PTPRC (CD45). Presence of at least one transcript of one of the markers or combinations of several markers defined the spot as leukocyte positive. We have added this information to the methods part and adapted the legend of Figure 1 accordingly.

In addition, we leverage the deconvolution algorithm Tangram for highlighting the diversity of cells within cytokine positive spots (Fig. S3). For these spots, we found a heterogeneous cellular composition, i.e. a mixture of T cells, innate cells, antigen presenting cells, epithelial and endothelial as well as B cells. However, this analysis confirmed our assumption that T cells were predicted to be present in each cytokine transcript-positive spot.

Figure S3

Figure S3: Tangram analysis of cytokine-transcript positive spatial spots reveals heterogeneous cellular compositions. Numbering of *IFNG* (A), *IL13* (C) and *IL17A* (E) transcript positive leukocyte spots in lichen planus, atopic dermatitis and psoriasis ST sections, respectively. The scale indicates the number of transcripts detected in each spot. Cellular composition of *IFNG* (B), *IL13* (D) and *IL17A* (F) transcript-positive spatial spots. The dotted line indicates T cells and innate lymphoid cells - the main producers of cytokines in human skin. For a better visualization, subtypes of cells were grouped together as T cells (cytotoxic-, helper-, regulatory T cells), innate cells (ILC1, ILC2, ILC3, NK), APC (dendritic cells, macrophages, mast cells, monocytes), epithelial cells (keratinocytes, fibroblasts, melanocytes), and endothelial cells (vascular-, lymphatic endothelial cells, pericytes).

Among the questions I could not answer from the manuscript or even a reading of the Supplemental Methods. These need to be front and center in the actual manuscript. - How were 'responder' genes defined? Is it the keratinocyte produced transcripts on provocation in lines 188-190 (table s6), well after extensive discussion of responder genes in the Results analysis? Or a different manual curation by the authors or an external method as implied in line 156? Are the genes responsive in terms of being directly induced genes, or simply also enriched in the Th subtype?

We thank the reviewer for pointing out this lack of clarity. For addressing this, we expanded the description in the supplemental methods, as well as referenced in manuscript and corresponding figure legend (now Figure S7) accordingly: " To evaluate the correlation between cytokine transcripts and

response induced in surrounding tissue cells, a responder gene signature to type 1, type 2, and type 3 mediated inflammation was developed by stimulating primary human keratinocytes *in vitro* with recombinant IFN-g, IL-13 and IL-17A (20 ng/ml each), respectively. After 16 hours, total RNA was isolated and whole genome expression arrays (SurePrint G3 Human GE 8X60K BeadChip (Agilent Technologies)) were performed according to the manufacturer's instructions. Gene expression data was filtered for p-value <0.05, adjusted p-value <0.05, and log₂ FC >1.5 for IL17A and IFNG or log₂FC >1 for IL-13. To further identify the most relevant responder genes *in vivo*, gene expression of cytokine transcript-positive spots residing in the epidermis were compared with the respective differentially expressed genes of *in vitro* stimulated keratinocytes (e.g. DEG of IL-17A+ spots with IL-17A stimulated keratinocytes). The overlapping gene signature as shown in Fig. S7 was then curated for genes being also present in the response signature of all other cytokines (e.g. IL-17A-specific genes in the IFN-g response signature and vice versa) IFNG positive spots/arrays). Hereby, 21, 29 and 7 responder genes could be identified for IL-17A, IFN-g and IL-13, respectively. As 3 out of the 7 IL-13 responder genes were also present in the IFN-g response signature, these genes were removed from the analysis and replaced by well-established genes to be directly induced by IL-13"

We adapted the figure legend accordingly and also added arrows and boxes to the figure to allow better understanding of the cytokine-specific response signatures. Using this procedure, we were able to analyze genes that are directly induced by the respective cytokines. However, a limitation of this is the restriction to the epidermis as dermal cells were not included in the in vitro stimulation. We have raised this limitation in the discussion as follows: "Our conditional density-based clustering method centers clusters around cytokine-positive spots, and consecutively optimises the radius of considered cytokine-specific responder signatures in each tissue slice according to in vitro stimulation of the epidermal keratinocytes (Supplemental Methods). This enabled us to calculate a spatial correlation, which can be generalised from epidermis to dermis when adjusting responder signatures, as well as to other diseases and tissues. Our method can be leveraged for identifying biomarkers and disease drivers in the future. By integrating three-dimensional spatial information using consecutive tissue sections, the algorithm could be further improved to identify disease-driving networks across tissue sections from the same patient, identifying antigen-specific T cell activation and may highlight promising precise treatment strategies."

- What are the full list of T cell marker genes used to call a spot positive for leukocytes? Were T cell subtypes considered?

As mentioned before, the full list of marker genes to determine a leukocyte positive spot was: CD2, CD3D, CD3E, CD3G, CD247 (CD3Z), and PTPRC (CD45). A spot was called positive if at least one transcript of one of these marker genes was present or a combination of several markers was detected. By this approach, we did not only acknowledge cytokine producing T cells, but also e.g. NK cells and innate lymphoid cells. Leukocytes were not subdivided into T cell subsets, but subdivided according to their cytokine expression. We have clarified this leukocyte identification strategy in the results ("To specifically focus on immune cells, spots were pre-sorted according to leukocyte markers (CD2, CD3D, CD3E, CD3G, CD247 (CD3Z), or PTPRC (CD45)). Presence of at least one UMI count of a single or combination of these markers was regarded as a leukocyte positive spot.") and methods section ("To characterize cytokine expressing cells, leukocytes were defined by expression of at least one of the marker genes CD2, CD3D, CD3E, CD3G, CD247 (CD3Z), and PTPRC (CD45) or combinations of these markers in the ST and single-cell datasets. Presence of one transcript was regarded as a positive association.")

- Do the authors believe a given T cell positive spot contained multiple T cells or just one? Does it matter? How would it affect the analysis if different types of T cells were found at a specific address?

As elaborated before, we have included more information on the spot size and the technique of spatial transcriptomics in general. Given a spot diameter of 55µM, it is quite likely that not only one cell resides in a spot and with a diameter of ~10µM also several T cells could be captured in this spot

area. We have investigated the cytokine transcript-positive spots also with the Tangram algorithm to elucidate their cellular composition, however, this prediction does not entirely indicate if one or more T cells of the same subtype are captured in a given spot (new Figure S3, see above). We also had a closer look into the distribution of spots being positive for two or more cytokines as this could identify spots with e.g. Th1 and Th2 cells (see figure for the reviewer below). Here, it could be shown that the majority of spots do only contain one cytokine, whereas double-positive spots represent the minority and triple-positive spots were not detected in the original data cohort.

So far, it cannot be ruled out that presence of one cell type does affect the activity and response of other cell types present in the same spot. To understand this fact better, deconvolution of spatial data can give further insights (see Tangram analysis above), however, to conclusively solve this, spatial transcriptomics needs to scale down the spot diameter. This may well be the next evolutionary step forward in spatial transcriptomics technologies.

Figure 1 for the reviewer and new Fig S1B

There is a major emphasis in the writing of this manuscript on there being a “single transcript” in a spot or T cell to the point that this is stated in the abstract. I am not sure this is even necessary to the most interesting point of the paper (numerous downstream target genes clustering spatially near Th cytokine detect). I found this assertion difficult to validate from the presented data and the approaches suspect, for example “Single-cell RNASeq analysis of psoriasis also indicated few transcripts per IFNG+ or IL-17A+ cell” on lines 125/126– are the authors using a notoriously shallow and low-sensitivity method (scRNA-seq) to validate their claim that detection of single transcripts is representative of the actual per cell expression?

We thank the reviewer for this important comment and agree that each technology has its limitations. We added an outlook to the discussion section to highlight the bottleneck of scRNA (“Further refinement needs to be implemented in terms of sensitivity of sequencing methods and the spatial resolution that with a spot diameter of 55µm captures more than just one cell. So far it cannot be ruled out that presence of one cell type does affect the activity and response of another cell type. Deconvolution of spatial spots using algorithms like Tangram (47) helps to understand the cellular composition of spots, however, a reduction of the spot diameter is highly desirable and may well be the next evolutionary step in spatial transcriptomics.”). In addition, we further emphasize that our hypothesis is intrinsically validated by spatial transcriptomics technology, i.e. we observe a strong spatial correlation of even low expressing cytokines with responder signatures in

close proximity. This is a strength of spatial transcriptomics data, which enabled this observation. The scRNA data is supportive of this hypothesis, and additional evidence. Although not fully understood yet, we believe that this transcriptional cascade initiated by a few cytokines may indeed be very important to reveal putative drug targets in the near future.

Immune response is spatially correlated with cytokine transcript number

If I understand this section, the authors are able to show that DEGs directly resulting from cytokine shifts (as per keratinocyte in vitro experiments) are found in spatial relation to addresses where that cytokine is detected. This is interesting and would have served more logically as the focal point of the paper. What is the resolution of these spatial hotspots compared with the hotspot of the technology? Are the differences distinct between diseases?

We agree with the reviewer that this is a very interesting observation. We show that response signatures are spatially correlated with cytokine-positive spots, which is the intrinsic validation of our hypothesis, i.e. few cytokine transcripts are enough to drive and maintain disease pathology. Regarding the resolution, this is somewhat limited by the fixed grid-size offered by the spatial transcriptomics technology. However, we thank the reviewer for this invaluable suggestion, and accordingly explored the radius as an additional free parameter in the density based clustering method (Fig. 5D). This has refined our novel clustering approach. We found the largest spatial correlation for radius of 0, 3, and 4 for IL17A, IL13 and IFNG, respectively (Fig. 5E). Notably, IL17A, IL13 and IFNG represent the hallmark cytokines for psoriasis, atopic dermatitis and lichen planus, respectively (Fig. 2Q). Interestingly, IFNG causes widespread inflammation, whilst IL17A and IL13 clearly decrease inflammation response over distance to cytokine-positive spots (Fig. 5E). This becomes evident by the suggested analysis of the reviewer. Furthermore, we have added Fig. 5D to visualize the impact of radiuses on our spatial density clustering method.

Figure 5: **D)** Representative tissue slice of psoriasis showing IL17A expression in relation to its responder signatures and different radiuses around the IL17A-positive leukocyte spot. The filling of each circle represents the UMI counts according to the scale on the right for either responder genes (green circle) or cytokine transcripts (blue circle for: IL17A). Red lines connect neighboring IL17A transcript-positive spots that together with the surrounding responder gene positive spots create a density-based cluster highlighted by a black line. **E)** Weighted Spearman correlation values for IFNG (orange), IL13

(red), and IL17A (blue) depending on the radius from the cytokine transcript-positive leukocyte spot. Strongest correlations for each cytokine are indicated with a circle.

Minor points

- It is really not optimal to apply a new large scale profiling approach like this on so few samples of each disease class. A single disease class would have been informative and the data much more interpretable.

We agree with the reviewer that more samples are always desirable. Focusing on one disease may have boosted statistical power, however, we also observed strong benefits in characterizing the diversity of inflammatory skin diseases, thereby, highlighting generalisability of our findings. In order to address the reviewer's comment, we have enlarged our patient cohort and included 16 new lesional samples (psoriasis n=6, atopic dermatitis n=4 and lichen n=6), which therefore doubles the size of the lesional biopsies. In addition, we also focussed on the common inflammatory skin diseases, i.e. lichen planus, atopic dermatitis and psoriasis and removed the three lesional PRP samples (still included in data release, but discarded from analysis). This resulted in 10.948 additional spots which were included across all analyses. Accordingly, we adapted figures and numbers throughout the whole manuscript, and are pleased that original findings reproduced. We believe this enlarged dataset strongly increases the value of our manuscript.

- Relatively limited information on clinical samples – PASI/EASI severity, lack of recent treatment, formal histopathology

We updated the material & methods section and included a supplemental table (Table S1) giving all the requested patient information.

- Line 108: “the tissue inflammation of nClSD is driven by T cell cytokines” – I believe that many researchers believe APCs and keratinocytes play a major role here.

We apologize for this inaccuracy and are in line with the reviewer that not only T cells drive skin inflammation. We therefore rephrased the sentence into: ‘As cytokines represent important drivers of tissue inflammation in nClSD’

Reviewer #2 (Remarks to the Author):

The manuscript by Schabitz and Hillig et al. characterize the spatial transcriptomic landscape of lichen planus, atopic dermatitis, psoriasis, and pityriasis rubra pilaris. They find general low cytokine expression from a few T cells that initiate a larger response in neighboring cells, which is validated using other sequencing methods, in situ hybridization, IHC, and flow cytometry. This is an interesting report that suggests active skin disease is maintained by a few active T cells. This work, along with the abundance of sequencing data, will be a valuable resource to the skin field once the data is publicly available. A few clarifications are needed to help the reader better understand the results as detailed below.

1) The sequencing data needs to be deposited in a publicly available database. The authors have boilerplate language that suggests this will be the case, but no specific accession number is provided.

We agree with the reviewer that FAIR (findable, accessible, interoperable and reusable) data is essential for open science, and have deposited our data on the GEO database. Currently the data was assigned a tracking number (NCBI tracking system #22992572) and as soon as we have the GEO accession number, we will add this information to the manuscript.

2) Going through the figures to understand the underlying data is a bit difficult given some of the necessary data is in supplementary figures. I suggest adding Figure S2A into the beginning of Figure 3 and Figure S4A into the bottom part of Figure 3. In addition, Figure 4D-F show representative cytokine expressers and their responder signatures in tissue slices. It would be helpful to denote where T cells are located to see if all T cells are active or only a subset within the tissue. Also, can the boundary between the epidermis and dermis be denoted by a line? It is difficult to determine where the boundary is with the superimposed spots.

We thank the reviewer for this suggestion and implemented it in the revised manuscript. As suggested, we included Fig. S2A at the beginning of Fig. 3. To further clarify that single cell data was used as a comparator to spatial data shown in Fig. 3, we removed the single cell data from Fig 3 and created a new Fig. 4. Here, we included, as suggested, the former Fig. S4A, and also Fig. S4C.

To follow the reviewer's suggestion, we also modified Figure 4 (now new Figure 5). Fig. 5 D,E emphasizes the developed density clustering algorithm with respect to cytokine actions in distinct radiuses from the cytokine transcript-positive spot (see outtake of the Figure 5 below). We furthermore deleted spots that did not contain cytokine- or responder gene transcripts for a better visualization and identification of the boundary between epidermis and dermis. By this, we have shown all spots being positive for IL-17A (blue circles) as representative example (Fig. 5 D). If the IL17A positive spot was neighbored by another one, the two spots were connected via a red line. The filled circle represents the presence of the responder gene signature highlighting that responder genes were highly expressed in close proximity to the cytokine positive spot, but not overall the tissue slice. To highlight these clusters of cytokine and responders, a black line was drawn around them. To not interfere with displayed clusters, we decided against denoting the boundary between the epidermis and dermis, as we think it is now much more visible with our applied changes. To clarify

this issue, we added the following text to the legend of the new Fig 5D: “The filling of each circle represents the UMI counts according to the scale on the right for either responder genes (yellow circle) or cytokine transcripts (blue circle for: IL17A). Red lines connect neighboring IL17A transcript-positive spots that together with the surrounding responder gene positive spots create a density-based cluster highlighted by a black line.”

Figure 5: D) Representative tissue slice of psoriasis showing IL17A expression in relation to its responder signatures and different radiuses around the IL17A-positive leukocyte spots. The filling of each circle represents the UMI counts according to the scale on the right capped at 800 UMI-counts for either responder genes (yellow circle) or cytokine transcripts (blue circle for: IL17A). Red lines connect neighboring IL17A transcript-positive spots that together with the surrounding responder gene positive spots create a density-based cluster highlighted by a black line. E) Weighted Spearman correlation values for IFNG (orange), IL13 (red), and IL17A (blue) depending on the radius from the cytokine transcript-positive leukocyte spot. Strongest correlations for each cytokine are indicated with a circle.

3) This figure legends could use more detailed information to describe what is shown in the figure panels. For instance, there are white and red circles in Figure 2F. The red circle is likely the blown up image, but I'm not sure what the white circles represent. There are also red lines connecting cells in Figure 4D and F and an orange line on one end of the conditional density cluster that remain a mystery.

We thank the reviewer for highlighting this ambiguity, and have increased the ‘readability’ of the figures as suggested. Here, we replaced the white circles in Fig. 2F by red ones that all indicate the presence of cytokine transcripts, and drew a line from the circle showing the enlarged image. The figure legend has been adapted accordingly.

Furthermore, we have enlarged information in the legend of Figure 4 (new Fig. 5) to clarify the red lines being a connection of cytokine positive spots (see above). The orange line was accidentally not colored in blue and revised in the figure and has now been removed.

4) Do the cytokine+ spots in the dermis also show positive correlation with responders?

This is an interesting suggestion, however, our developed responder signature was retrieved from *in vitro* stimulated keratinocytes, therefore, it unfortunately does not allow us to use it for dermal spots containing fibroblasts. Nevertheless, this is a valuable suggestion to be followed up in the future. We added this as outlook in the discussion: “Our conditional density-based clustering method centers clusters around cytokine positive spots, and consecutively optimises the radius of considered cytokine-specific responder signatures in each tissue slice according to *in vitro* stimulation of the epidermis (Supplemental Methods). This enabled us to calculate a spatial correlation, which can be generalised from epidermis to dermis when adjusting responder signatures, as well as to other diseases and tissues.”

Reviewer #3 (Remarks to the Author):

The authors have conducted an impressive analysis of spatial sequencing data from a group of inflammatory skin diseases. They found that a relatively small number of T cell cytokine producing cells are present in the epidermis and upper layers of the dermis that distinguish different diseases. Importantly, the authors have developed a novel bioinformatics approach to identify cytokine responder genes in the adjacent spatial landscape.

1. Impact of the manuscript.

a. The author state in the abstract that “Despite the expected T cell infiltration, we observed a rather low frequency of only 1-10 pathogenic T cell cytokine transcripts per skin section.” This appears to have been reported previously using in situ hybridization (see Comment #4) in psoriasis and atopic dermatitis.

We agree with the review that this has been reported before. Please see answer to comment #4, which is comprehensively addressing this valid concern.

b. The others further state: “Nevertheless, cytokine expression was limited to lesional skin, presented in a disease-specific pattern and evoked specific responder signatures in direct proximity of cytokines. This showed that single cytokine transcripts initiate amplification cascades of thousands of specific responder transcripts forming localized epidermal clusters. Thus, within the abundant and heterogeneous T cell infiltrates of ncISD, only a few T cells drive disease by initiating an inflammatory amplification cascade in their local microenvironment.” It is certainly known that T cell cytokines act in trans- for example IFN-gamma and IL-17 have downstream effects on keratinocytes. However, the authors developed a novel approach to study the downstream effects of T cell cytokines in adjacent cells by density-based clustering. This methodological advance is significant and should be emphasized. The authors leverage this approach to demonstrate that T cell cytokine downstream responder genes are enriched in adjacent spots. Is it possible to use this approach to identify novel genes associated with the T cell cytokines that are possible downstream responder targets?

We thank the reviewer for the suggestion to emphasize our novel density-based cluster approach, which we have further highlighted throughout the manuscript accordingly. We are convinced that our method allows us to understand the functional relevance of cytokines and other secreted factors in more depth.

The suggestion of the reviewer to data-driven identify unknown cytokine responder genes based on the spatial transcriptomic data and our density-based clustering is intriguing. For this, we performed a differential gene expression analysis of density clusters using solely the cytokine+ spots without responder spots versus remaining spots in the epidermis. For IL17A we found 224 responder genes requiring a cut-off of $\log_2FC > 1$ and $padj\text{-value} < 0.001$, for IFNG we got 185 responder genes, and for IL13 we identified 13 responder genes (new Fig. S8A-B). By this, some yet unknown cytokine gene associations such as CRABP2 and APOL1, GBP1, WARS1, SRGN, SERPINB1, LYZ, HLA-DRB1, RAC2, and CCL17, KRT6C, CLEC10A were identified for IL17A, IFNG and IL13, respectively.

Figure S9

Figure S8: Spatial transcriptomics enables to identify new cytokine responder genes and thereby potential drug targets **A-C)** Differentially expressed genes were analysed in radial proximity (radius 1-5) to the cytokine transcript-positive spot (radius 0). Numbers of data derived responder genes for each radius are shown for *IFNG* (A), *IL13* (B) and *IL17A*(C). Experimentally identified responder genes (gold or grey (non significant) and data derived responder genes (up-regulated in blue; down-regulated in red) for *IFNG* in radius 4 (D), *IL13* in radius 5 (E) and *IL17A* in radius 1 (F).

We agree with the reviewer that the amplification cascade of cytokines has been reported before, and acknowledge this in the manuscript and abstract accordingly now. Novelty is that we have experimental sensitivity to detect cytokines, which distinguish disease-driving from bystander T cells, which is confirmed with spatial response pattern and our method. Now we highlight this novel contribution more: “Abundant heterogeneous immune cells infiltrate chronic inflammatory diseases and characterization of these cells is needed to distinguish disease-driving from bystander immune cells. Here, we investigated the landscape of non-communicable inflammatory skin diseases by spatial transcriptomics resulting in a large repository of 62,000 spatially defined human cutaneous transcriptomes of 31 patients. Despite the expected immune cell infiltration, we observed a rather low frequency of pathogenic disease-driving cytokine transcripts per skin section. Nevertheless, cytokine expression was limited to lesional skin and presented in a disease-specific pattern. Leveraging a density-based spatial clustering method, we identified specific responder signatures in direct proximity of cytokines, and confirmed that single cytokine transcripts initiate amplification cascades of thousands of specific responder transcripts forming localized epidermal clusters. Thus, within the abundant and polyclonal T cell infiltrates of ncISD, only a few T cells drive disease by initiating an inflammatory amplification cascade in their local microenvironment.”

2. UMIs vs. spots. The authors report “Despite the expected T cell infiltration, we observed a rather low frequency of only 1-10 pathogenic T cell cytokine transcripts per skin section.” They further state “Taking the whole section into account, we detected only a few transcripts for IFNG, IL13 or IL17A (272, 57, or 92 UMI counts in all sections, respectively) in lesional skin (Fig. 2E).” There were 18 lesional samples, but we don’t know the distribution of the UMI counts for each sample as well as how many

spots were detected with the distribution of UMIs per spot. Should these data be normalised for the size of the biopsy as they are counting the entire section which may vary, or perhaps they were of uniform size? What is the variability of the data? Are the majority of UMIs coming from 1-2 samples?

We agree with the reviewer that more in depth information needs to be given on presence of spots per section and UMI counts within these spots. We therefore included a new supplemental table (Table S2) giving detailed information on each single patient, all sections and UMI counts of each cytokine. This table also elucidates that presence of cytokines is not distributed equally over the entire dataset, but is specific for the diseases investigated (also shown in Fig. 2N-Q). The mean number of spots per section is 767 ± 293 . Here, lesional skin is represented by significantly more spots per section (823 ± 324 vs 633 ± 125 , $p=0.0015$) which reflects morphological changes in the tissue due to inflammation. This is further supported by the UMI counts per spot per section being 3189 ± 6620 in lesional skin and 605 ± 613 in non-lesional skin ($p=0.0002$). We added this information to the methods section of the manuscript.

In our analysis pipeline, only lesional samples were included and due to the low variation in spots per section, data was not normalized to the size of biopsy.

We can infer the minimum number of cells by the number of spots. If there is one UMI in a spot, then there is one cell, if more UMIs per spot, there could be more cells but there is at least 1.

The reviewer is right. If there is just one UMI count detected, there needs to be at least one cell in this specific spot. This, however, does not necessarily mean that two UMI counts mean two cells. To get an overview on the cellular composition of each spot, we ran the Tangram algorithm (Biancalani et al., Nat Methods. 2021) to deconvolute cytokine producing spots. This data is included in the new Fig. S3 and showed that cytokine producing T cells and ILCs are present in each cytokine transcript-positive spot, however in varying percentages ranging from a few percent to up to 60%.

Figure S3

Figure S3: Tangram analysis of cytokine-transcript positive spatial spots reveals heterogeneous cellular compositions. Numbering of *IFNG* (A), *IL13* (C) and *IL17A* (E) transcript positive leukocyte spots in lichen planus, atopic dermatitis and psoriasis ST sections, respectively. The scale indicates the number of transcripts detected in each spot. Cellular composition of *IFNG* (B), *IL13* (D) and *IL17A* (F) transcript-positive spatial spots. The dotted line indicates T cells and innate lymphoid cells - the main producers of cytokines in human skin. For a better visualization, subtypes of cells were grouped together as T cells (cytotoxic-, helper-, regulatory T cells), innate cells (ILC1, ILC2, ILC3, NK), APC (dendritic cells, macrophages, mast cells, monocytes), epithelial cells (keratinocytes, fibroblasts, melanocytes), and endothelial cells (vascular-, lymphatic endothelial cells, pericytes).

3. Validation by in situ hybridization. The authors state for their in situ hybridization validation that “The median number of transcript-positive cells per section for *IFNG*, *IL13*, and *IL-17A* mRNA were 16, 2, and 0 for LP, AD, and psoriasis, respectively, thus confirming our observations from the ST analysis (Fig. 2G).” They also provide similar data for mean UMI counts per biopsy for scRNA-seq data. Should this be normalised to the size of the section? But this cannot be directly compared to the spatial sequencing data. Also, why is the mean for *IL17A* “0” in psoriasis? Isn’t this a Th17 disease? Its not clear what the above numbers exactly refer to.

We thank the reviewer for highlighting this inaccuracy. We calculated the mean values for *IL17A*, *IL13* and *IFNG* across all sections of all different diseases. As the reviewer pointed out, *IL17A* is mostly

expressed in psoriasis and to a lesser extent in LP. Therefore, we now calculated the mean transcript number of each cytokine per disease delivering 11 *IL17A* transcript-positive cells per psoriasis section, 83 *IFNG* transcript-positive cells in LP and 4 *IL13* transcript-positive cells in AD.

These numbers were included in the results: "The median number of transcript-positive cells per section for *IFNG*, *IL13*, and *IL-17A* mRNA were 83, 4 and 11 for LP, AD, and psoriasis, respectively, thus confirming our observations from the ST analysis (Fig. 2G)."

4. In situ hybridization as compared to the literature. I easily found two relevant articles that were not quoted (see below), and it would seem that they find more cytokine+ cells, albeit it is hard to compare as the calculations are different. Is it possible that there is dropout in the authors' data?
- JID Innovations, Volume 1, Issue 2, June 2021, 100021, Alice Wang et al, JID Innovations, Cytokine RNA In Situ Hybridization Permits Individualized Molecular Phenotyping in Biopsies of Psoriasis and Atopic Dermatitis
 - British Journal of Dermatology, Volume: 185, Pages: 585-594. Signalling of multiple interleukin (IL)-17 family cytokines via IL-17 receptor A drives psoriasis-related inflammatory pathways. M.A.X. Tollenaere, J.

We thank the reviewer for this note. Indeed, both publications support our findings: cytokine mRNA positive cells are predominantly present in the epidermis, they have a low frequency, but are expressed in a disease-specific manner. In addition, both publications highlight that cytokine expression occurs mainly in CD3+ T cells. We agree with the reviewer that both publications tend to show higher frequencies of cytokine mRNA positive cells, however, both publications also indicate a high heterogeneity amongst investigated patients. Furthermore, ISH techniques and analysis methods were different and could explain the observed deviation. Nevertheless, both publications support our findings and now are referenced in the discussion as follows "These findings are supported by recent publications investigating cytokine mRNA positive cells in inflamed skin highlighting a predominant expression in the epidermis and a co-localization with CD3 expression (44, 45)."

5. Responder genes-specificity. In Fig 3B for example, the authors show that *IL17A* responder genes are enriched in *IL17A*+ spots, but many genes are unlabeled. The specificity of this response would be made clear by showing that *IFN-gamma* and *IL-13* responder genes were not enriched. Also, if the detection of cytokine and responder genes is robust, shouldn't the key cytokine receptor genes be detected as well? The genes encoded receptors for *IFN-gamma*, *IL-13*, *IL17A* should be detected in Fig. 4D, E, F. Also, there is no information on the cell types found in cytokine+ spots or the adjacent responder spots, since each spot will be a mixture of cells.

*We thank the reviewer for this criticism and included some more data to solve it. The volcano plot given in Fig. 3B (now Fig. 3C) indeed shows leukocyte related genes (purple) and *IL-17* responder genes (blue). Furthermore, we highlighted *IL-17* receptors in Fig. 3C reflecting the potential of keratinocytes to respond to the presence of the cytokine and also added this information on *IFNG* as well as the *IL13* receptors in Figure S4A and C, respectively. However, the receptors were all expressed below the statistical threshold.*

As cytokine receptors are usually difficult to detect by RNASeq, we did not include the respective receptors into the response signature for each cytokine. Therefore, these receptors are not part of the clusters shown in Fig. 5F-H. However, as described before, we have refined our cluster algorithm

and thereby are now also able to identify yet unknown associations between a cytokine and its tissue response signature (new Fig. S8) and have analysed cytokine positive spots for their cellular composition using the Tangram algorithm.

6. Validation by scRNA-seq. “Confirming the spatial dataset, most leukocyte-associated genes were also identified in the single-cell data, whereas the responder gene signatures were widely missing in the single-cell data (Fig. 3F, Fig. S4C, D).” It is not clear which responder genes are missing in the single-cell data to assess how widely they are missing. Can imputation be used to demonstrate this?

We apologize for our imprecise wording and replaced the phrase ‘widely missing responder gene signature’ by ‘Confirming the spatial dataset, all leukocyte-associated genes were also identified in the single-cell data within the leukocyte and antigen-presenting cell cluster, whereas the response gene signatures for each cytokine were either not detected or found below statistical significance (Fig. 4C; Fig. S4) highlighting their specificity for epidermal keratinocytes.’

Also, the authors previously stated “T cell genes associated with IL17A were IL17F, IL22, and IL26, and the responder signature of IL17A consisted of e.g., IL19, NOS2, S100A7A, DEFB4A, CXCL8, and IL36G (Fig. 3B, C)”, but are showing different genes in Fig. 3F.

We thank the reviewer for this comment. In the old Fig. 3F, IL17A transcript-positive leukocyte spots from the spatial data set were compared with IL17A transcript-positive leukocytes from the single-cell data set. As the spatial IL17A transcript-positive leukocyte spots were residing in the epidermis, they also contained keratinocyte-derived responder genes, whereas the single-cell IL17A transcript-positive leukocytes did not. Therefore, the spatially derived responder genes were either present in the non-significant fraction of the correlation plot given in the old Fig. 3F, or could not be traced at all. Due to the latter case, the number of denoted genes varied between Fig. 3B, C and Fig. 3F.

As suggested by reviewer 1, we increased the size of the spatial cohort. By this the already low correlation of IL17A transcript-positive spatial spots and IL-17A transcript-positive single cells decreased from $r=0.34$; $p=1.5e-03$ to $r=0.02$; $p=9.31e-01$. In addition, the corresponding correlation of IFN γ decreased from $r=0.52$; $p=2.16e-04$ to $r=0.39$; $p=6.28e-03$. We therefore decided to remove these misleading graphs from the manuscript.

To enhance the comparability of both analyses (spatial and single-cell), we added all genes shown in the spatial analysis in Fig. 3B (now Fig 3C) also in Fig. 4C and consequently also in Fig. S4 for IL13 and Fig. S4 and S5 for IFN γ and if those genes could be detected as well in the single cell dataset. We have revised the respective part in the results section as described above. To further prevent confusion on spatial and single cell data sets, we moved the single cell data into a new Fig. 4.

7. Leukocytes. “To characterize cytokine expressing cells, leukocytes were defined by the marker genes CD2, CD3D, CD3E, CD3G, CD247, and PTPRC in the ST and single-cell datasets. Leukocytes were defined as cytokine-positive if at least one UMI-count of the cytokine gene was detected.” Aren’t these T cells?

By the combination of the above described markers, we were not only retrieving spots containing T cells, but also spots with innate lymphoid cells and NK cells that are also capable of producing cytokines. To clarify this fact in the manuscript, we have added the more common gene names for

PTPRC, CD45, and for CD247, CD3Z, in the legend of Figure 1 and the methods section also clarified that presence of only one marker gene or the combination of several markers classified the spot or cell as a leukocyte.

8. The authors state “To characterize cytokine expressing cells, leukocytes were defined by the marker genes CD2, CD3D, CD3E, CD3G, CD247, and PTPRC in the ST and single-cell datasets.” It is not clear what quantitative criteria was used for this definition.

We apologize for the inaccuracy. In order to define a cell, for single-cell data, or a spot, for ST data, as leukocyte we required that at least 1 transcript of any of these marker genes in a cell or spot was measured. This means that a cell or spot can contain only one of the above named marker genes as well as any combination. we have clarified this in the respective methods section: “To characterize cytokine expressing cells, leukocytes were defined by expression of least one of the marker genes CD2, CD3D, CD3E, CD3G, CD247 (CD3Z), and PTPRC (CD45) or combinations of these markers in the ST and single-cell datasets. Presence of one transcript was regarded as an expression.”

REVIEWER COMMENTS

Reviewer #1 (Remarks to the Author):

The revisions by the authors substantially improve the technical explanation of the study.

What data from this study shows convincingly is that at the resolution of Visium, i.e. 55 μM , that spots with higher cytokine expression correlate with significantly higher transcription of a set of genes. It is very likely that most but not all of what the authors call 'driver cytokines' are produced by T cells, and that most but not all of what the authors term 'response genes' are produced by keratinocytes. The keratinocyte response profiling was important. Therefore, it is credible and novel that local immune microenvironments in chronic skin disease are detected by this study, an assertion that is supported by the layer and disease-specificity of cytokine expression.

The attempt to quantify the transcript levels as surprisingly sparse is problematic and not supported by my reading of the data. Even with the toned-down language in the revision, the abstract includes the statements "a rather low frequency of pathogenic disease driving cytokine transcripts per disease section" and "only a few immune cells drive disease by initiating an inflammatory amplification cascade in their local microenvironment". Either 1) the sensitivity of the assay is low, which is undoubtedly true at this juncture in the technology or if the frequency correlates to bulk RNA sequencing results and flow presented here, which correspond to past work in the field, then 2) the numbers of cytokine-positive T cells is a subtraction, which was already known. What is the "high" comparator of "low" – is it allergic contact dermatitis or the number of transcripts in inflamed circulating T cells?

Major points

1. Page 7: The authors comment "Due to the size of every spot ($\text{\O}55\mu\text{m}$), DEG generally displayed genes derived from cytokine producing cells, so called driver genes, and genes originated from cytokine responding cells in close proximity, so called responder genes." There is an overarching issue that the DEGs are presumed first to be responder genes and then on Page 9 the in vitro keratinocyte DEGs are matched. The authors should call the DEGs DEGs, and after defining in vitro "response" genes, specifically measure their overall statistical overrepresentation as a whole in the DEGs in an unbiased analysis.

2. Regarding comment 1, this lack of clarity plays out in instances such as Page 10: "a $\log_2\text{FC}$ cut-off of > 1 and $\text{padj-value} < 0.001$, we identified 224 responder gene transcripts for IL17A..." this really means DEGs I believe? How many were in vitro responders etc? The terminology should be more crisply defined and the sets formally statistically compared – it may be entirely possible it strengthens the authors argument. For example in the methods it is stated "The overlapping gene signature...was then curated for genes being also present in the response signature of all other cytokines (e.g. IL-17A-specific genes in the IFN- γ response signature and vice versa) Hereby, 21, 29 and 7 responder genes could be identified for IL-17A, IFN- γ and IL-13, respectively. Were these...overrepresented in the DEGs or could this be false discovery?"

3. In the single sentence summary and elsewhere, it is stated "Single specific cytokine transcripts initiate local inflammatory amplification cascades in inflammatory skin diseases." What is amplified? Is the amplification simply = many responder genes in keratinocytes to a single driver cytokine? As in other places in this manuscript I have concerns that a concept that is not really established is being presented as shorthand.

Minor points

1. The Visium technology is not even mentioned by name in the Results section. At least 3-4 sentences in the Methods should be pulled into Results.

Reviewer #2 (Remarks to the Author):

The revised manuscript addresses my main critiques and their response has strengthened the manuscript. This work should be a great resource to the field and I recommend acceptance.

Reviewer #3 (Remarks to the Author):

1. Impact of the manuscript. The authors appropriately revised the manuscript to reflect areas in which their findings have already been reported by others. In response to the suggest that they may use their novel density-based cluster approach to identify new downstream targets of various cytokines they report methodology "Identification of potential new responder genes" and note, for example that APOL1, GBP1, WARS1, SRGN, SERPINB1, LYZ, HLA-DRB1, RAC2 are novel downstream targets of IFNG. It is not clear how they arrived at this conclusion. A cursory search showed that GBP1 and HLA-DRB1 are downstream targets of the IFNG Reactome pathway (<https://reactome.org>) and Enricher indicates that SERPINB1, APOL1, GBP1 and HLA-DRB1 are upregulated in interferon- γ human keratinocyte GDS4601. WARS1, HLA-DRB1 were listed as part of the MSigDB Hallmark 2020 interferon gamma response. And this was a limited search. Although this reviewer thought that the technique could be used to identify novel downstream targets, the methodology to do an exhaustive search of databases may be quite difficult. The authors summarize the impact of their manuscript as "Novelty is that we have experimental sensitivity to detect cytokines, which distinguish disease-driving from bystander T cells, which is confirmed with spatial response pattern and our method."

2. UMIs vs. spots. The authors present their data in Supplemental Table 2. As the authors state "This table also elucidates that presence of cytokines is not distributed equally over the entire dataset, but is specific for the diseases investigated (also shown in Fig. 2N-Q). The mean number of spots per section is 767 ± 293 ." IL-13 is robustly expressed in one atopic dermatitis sample. The next highest expression is half of the most robust at best, but this tissue has 2-3 times the number of spots, so the overall expression is even less. I am not sure if there is a statistical methods to account for the variability across samples. With apologies, I do not know that Tangram method that the authors report.

3. Validation by in situ hybridization. The authors have confirmed cytokine expression by in situ hybridization but we don't know if there is variation and whether the differences are significant.

4. In situ hybridization as compared to the literature. It is appropriate that the authors are referencing the literature.

5. Responder genes-specificity. The authors have revised the figure to include additional genes.

6. Validation by scRNA-seq. The authors have included two Figure 4s?

7 and 8. Leukocytes. I am not clear how NK cells were defined.

A minor point:

in revised Figure S3B - the epithelial cell signatures are a minority of the signal in the epidermis, which doesn't make much sense.....and in revised Figure S3D the epithelial component is larger in the dermis than the epidermis..

REVIEWER COMMENTS

Reviewer #1 (Remarks to the Author):

The revisions by the authors substantially improve the technical explanation of the study.

What data from this study shows convincingly is that at the resolution of Visium, i.e. 55 μM , that spots with higher cytokine expression correlate with significantly higher transcription of a set of genes. It is very likely that most but not all of what the authors call 'driver cytokines' are produced by T cells, and that most but not all of what the authors term 'response genes' are produced by keratinocytes. The keratinocyte response profiling was important. Therefore, it is credible and novel that local immune microenvironments in chronic skin disease are detected by this study, an assertion that is supported by the layer and disease-specificity of cytokine expression.

We thank the reviewer for the kind feedback and for acknowledging the novelty of our results.

The attempt to quantify the transcript levels as surprisingly sparse is problematic and not supported by my reading of the data. Even with the toned-down language in the revision, the abstract includes the statements "a rather low frequency of pathogenic disease driving cytokine transcripts per disease section" and "only a few immune cells drive disease by initiating an inflammatory amplification cascade in their local microenvironment". Either 1) the sensitivity of the assay is low, which is undoubtedly true at this juncture in the technology ...

We thank the reviewer for raising this point, thereby giving us the opportunity to further clarify. Spatial transcriptomics is a new technology and, as every new technique, has challenges which need to be overcome. In order to address this, we have extended the discussion further on the limitations and methodical challenges of spatial transcriptomics:

"However, it is clear that ST is subjected to methodical challenges. Further refinement needs to be implemented in terms of the spatial resolution that with a distance from the neighboring spot center-to-center of 100 μm is omitting valuable information and with a spot diameter of 55 μm captures more than just one cell."

For a balanced evaluation, we also highlight technical advantages of spatial transcriptomics in the discussion now:

"Preserving spatial information, while being independent of long digestion steps, is enormously beneficial in tissue systems like skin with distinguishable functional units. In essence, ST enables researchers to investigate whole transcriptome sequencing data in the context of interacting units in complex tissues."

In addition, we further toned down the abstract and specified the statements we make:

“Leveraging a density-based spatial clustering method, we identified specific responder signatures in direct proximity of cytokines, and confirmed that detected ~~single~~ cytokine transcripts initiate amplification cascades (...).”

We also quantified ambiguous terms such as “...rather low frequencies of pathogenic disease driving cytokine transcripts.”

“Despite the expected immune cell infiltration, we observed rather low numbers of pathogenic disease driving cytokine transcripts (IFNG, IL13 and IL17A), i.e. >125 times less compared to the mean expression of all other genes over lesional skin sections.”

It is our strong belief that spatial transcriptomics offers unique research opportunities, here exemplified, spatial resolution of inflammatory skin diseases empowered to correlate expression of a cytokine and its direct response in close proximity, which the reviewer kindly acknowledged as novelty. Thereby, our method and analysis inherently proves that low expressing cytokines are indeed driving the response, which is evident by the observed correlations, and consequently rules out the concern that the sensitivity of spatial transcriptomics would be too low to detect cytokines. To clarify, we further elaborate this point in discussion:

“This enabled us to analyse the impact of detected transcripts on their direct surroundings forming local immune microenvironments and calculate a distinct spatial correlation, independent of sample size or heterogeneous cytokine distribution.”

At first, we were actually surprised to see the low frequency of cytokine UMI counts in inflamed skin and were also wondering if this could be due to technical issues. Therefore, in situ hybridization was performed delivering comparable results and further supporting our spatial findings. All scientific evidence collected through analysis and additional validation experiments point to the same conclusion, that we indeed have the sensitivity to reliably quantify cytokine expression in spatial transcriptomics. To make this clear, we further highlight the validation experiments as supporting evidence in the result and discussion section:

“Being a technology with extended resolution properties, we additionally performed in situ hybridisation to support our ST analysis and, by delivering comparable results, confirmed our central findings.”

In summary, the strong correlation of cytokine transcripts with responder genes in close proximity and additional experimental validation thereby conclusively show the functionality of just a few transcripts in the tissue. Despite the limitation of spatial transcriptomics, this new technology allows deep insights into the architecture of skin inflammation (see discussion):

“Even though further reduction of the spot diameter is highly desirable and may well be the next evolutionary step in ST, our analysis shows that exploiting the capabilities of ST to the fullest offers unique research opportunities and empowers to investigate the architecture of skin inflammation.”

... or if the frequency correlates to bulk RNA sequencing results and flow presented here, which correspond to past work in the field...

Thanks for this comment. In line with past work in the field, our FACS data shows that ~10% of T-cells derived from lesional psoriasis skin are capable of IL17A production after in vitro stimulation with PMA/ Ionomycin. Notably, neither bulk nor single cell sequencing proves that a few cytokine transcripts drive inflammation, although pointing to this, and may be a likely hypothesis. In contrast, the spatial component empowers to conclusively illustrate that few cytokine transcripts indeed drive the inflammation by investigating responder gene signatures in close proximity. The observed correlation between cytokines and responder genes clearly highlights this functional regulation, whilst only blurry depicted in bulk or single cell sequencing. Above all, this approach is conducted without harsh dissociation procedures or stimulation of the investigated tissue. To clarify this, we have refined the discussion:

"Bulk and single cell sequencing of lesioned skin suggested that few cytokines may drive inflammation, and this is further strengthened by the observed correlation between cytokine transcripts and responder genes in spatial context, thus giving further evidence of functional regulation."

..., then 2) the numbers of cytokine-positive T cells is a subfraction, which was already known.

We agree with the reviewer that our finding is concordant with established beliefs / knowledge, i.e only a subfraction of T cells actively participates in inflammatory processes, and that most of the T cells infiltrating the tissue are somewhat bystanders. So far, however, we were not able to investigate the function of these few, disease-relevant cells in depth, which was recently enabled by spatial transcriptomics (see detailed responses above).

What is the “high” comparator of “low” – is it allergic contact dermatitis or the number of transcripts in inflamed circulating T cells?

*We apologise for ambiguous wording of ‘low’ and ‘high’ expressed genes, which needs to be quantified and put in context. For this, we investigated the mean expression of genes over a section, and specifically compared cytokine transcripts versus other genes (see boxplot below, added to **Figure S1C**). This highlights that the expression of cytokine transcripts is approximately 7 $\log_2(\text{fold changes})$ smaller compared to other genes ($p\text{-value} = 1.40\text{E-}02$) in lesional skin and 10 $\log_2(\text{fold changes})$ smaller compared to other genes ($p\text{-value}=2.2\text{E-}3$) in non-lesional skin. Furthermore, we have added the relation of cytokine transcripts and responder genes to **Figure S7H** highlighting that responder gene transcript are 270 time higher expressed than their inducing cytokine transcripts ($p\text{-value}=5.71\text{E-}3$)*

Fig. S1C. Cytokine transcripts are approximately 7 $\log_2(\text{fold change})$, i.e. >125 times, lower expressed compared to other genes.

Fig. S7H. Cytokine transcripts are approximately 8 $\log_2(\text{fold change})$, i.e. >270 times, lower expressed compared to their respective responder genes.

In addition, we report this $\log_2(\text{fold change})$ in the abstract now:

“Despite the expected immune cell infiltration, we observed rather low numbers of pathogenic disease driving cytokine transcripts (IFNG, IL13 and IL17A), i.e. >125 times less compared to the mean expression of all other genes over lesional skin sections.”

Major

points

1. Page 7: The authors comment “Due to the size of every spot ($\text{\O}55\mu\text{m}$), DEG generally displayed genes derived from cytokine producing cells, so called driver genes, and genes originated from cytokine responding cells in close proximity, so called responder genes.” There is an overarching issue that the DEGs are presumed first to be responder genes and then on Page 9 the in vitro keratinocyte DEGs are matched. The authors should call the DEGs DEGs, and after defining in vitro “response” genes, specifically measure their overall statistical overrepresentation as a whole in the DEGs in an unbiased analysis.

We thank the reviewer for highlighting the confusing nomenclature used throughout the manuscript. We have reviewed the entire manuscript and have revised the nomenclature as the

reviewer suggested: As long as responder genes have not been defined by using stimulated keratinocytes (Fig. S7), regulated genes are named as 'DEGs'. Just for our density cluster algorithm, cytokine regulated genes have been termed 'responder genes'.

To further exclude that the identified responder genes are overrepresented in the dataset, we performed an enrichment analysis showing that responder genes follow the overall gene expression in cytokine positive spots. We have added this information now to Fig. S7 and show the excerpt here for the reviewer:

Figure S7E-G: Enrichment analysis of response signature genes for E) IL17A, F) IFNG, and G) IL13 (black bars) within the DEG of the respective cytokine transcript-positive spots.

2. Regarding comment 1, this lack of clarity plays out in instances such as Page 10: “a log₂FC cut-off of > 1 and padj-value < 0.001, we identified 224 responder gene transcripts for IL17A...” this really means DEGs I believe? How many were in vitro responders etc? The terminology should be more crisply defined and the sets formally statistically compared – it may be entirely possible it strengthens the authors argument. For example in the methods it is stated “The overlapping gene signature...was then curated for genes being also present in the response signature of all other cytokines (e.g. IL-17A-specific genes in the IFN- γ response signature and vice versa) Hereby, 21, 29 and 7 responder genes could be identified for IL-17A, IFN- γ and IL-13, respectively. Were these...overrepresented in the DEGs or could this be false discovery?”

Please see the response above, we have clarified the DEG/responder gene issue throughout the manuscript. In particular, we have clarified that we have determined cytokine-related expression of genes and not necessarily responder genes.

“With a log₂FC cut-off of > 1 and padj-value < 0.05, we thereby identified 974 IFNG-related, 148 IL13-related, and 228 IL17A-related upregulated DEGs (Fig. S8A-C).”

In addition, we double checked all responder signatures and can confirm that only four IL-13-specific genes were discarded because of the curation process, and highlighted in method section accordingly:

“...was then curated for genes being present in the response signature of all cytokines (e.g. IL-17A-specific genes in the IFN- γ response signature and vice versa). This, however, did only lead to exclusion of four IL-13-specific genes that were also present in the IFN- γ response signature.”

In general, all analyses performed throughout the manuscript were non-hierarchical, and therefore, overrepresentation of results should not occur. This is further supported by an enrichment plot showing the overall up-regulated genes in a cytokine transcript-positive spot along a gaussian distribution which is followed by the expression of our selected response genes (see Fig. S7E-G above).

3. In the single sentence summary and elsewhere, it is stated “Single specific cytokine transcripts initiate local inflammatory amplification cascades in inflammatory skin diseases.” What is amplified? Is the amplification simply = many responder genes in keratinocytes to a single driver cytokine? As in other places in this manuscript I have concerns that a concept that is not really established is being presented as shorthand.

We thank the reviewer for this comment. Indeed, we introduce a new concept in just a few words. To explain the concept in more depth, we have added the following paragraph to results and discussion:

“Strikingly, the few cytokine positive spots having only 1 to 15 (IFNG: 1 to 8, IL13: 1 to 3, IL17A: 1 to 15 UMI counts/spot) cytokine transcripts were able to induce up to 25,000 responder transcripts in the surrounding spots indicating a tremendous amplification of the cytokine signal and thereby an amplification of tissue inflammation.”

In addition, we provided a supplemented boxplot quantifying the low and high expression (see above, Fig. S1C and Fig. S7H).

Minor points

1. The Visium technology is not even mentioned by name in the Results section. At least 3-4 sentences in the Methods should be pulled into Results.

We thank the reviewer for this suggestion, and have extended our objective paragraph at the end of introduction:

“Here, we investigated adaptive immune responses in lesional and non-lesional skin of ncISD with spatial resolution using the Visium technology of 10X Genomics.”

and we expanded the result section:

“Gene expression was measured in frozen and H&E stained skin sections using the Visium technology of 10X Genomics.”

Reviewer #2 (Remarks to the Author):

The revised manuscript addresses my main critiques and their response has strengthened the manuscript. This work should be a great resource to the field and I recommend acceptance.

We thank reviewer #2 for the kind evaluation.

Reviewer #3 (Remarks to the Author):

1. Impact of the manuscript. The authors appropriately revised the manuscript to reflect areas in which their findings have already been reported by others. In response to the suggest that they may use their novel density-based cluster approach to identify new downstream targets of various cytokines they report methodology “Identification of potential new responder genes” and note, for example that APOL1, GBP1, WARS1, SRGN, SERPINB1, LYZ, HLA-DRB1, RAC2 are novel downstream targets of IFNG. It is not clear how they arrived at this conclusion. A cursory search showed that GBP1 and HLA-DRB1 are downstream targets of the IFNG Reactome pathway (<https://reactome.org>) and Enricher indicates that SERPINB1, APOL1, GBP1 and HLA-DRB1 are upregulated in interferon- γ human keratinocyte GDS4601. WARS1, HLA-DRB1 were listed as part of the MSigDB Hallmark 2020 interferon gamma response. And this was a limited search. Although this reviewer thought that the technique could be used to identify novel downstream targets, the methodology to do an exhaustive search of databases may be quite difficult. The authors summarize the impact of their manuscript as “Novelty is that we have experimental sensitivity to detect cytokines, which distinguish disease-driving from bystander T cells, which is confirmed with spatial response pattern and our method.”

We agree with the reviewer that calling them “novel responder genes” is rather misleading, since this is only a data-driven expansion of putative responder genes. In order to address this, we have toned down our wording accordingly (see results):

“By this, we data-driven expanded our definition of cytokine-gene associations such as SRGN, LYZ and CCL17, CLEC10A and GM2A for IFNG, IL13 and IL17A, respectively (Fig. S8).”

Furthermore, in the discussion we raise that additional experimental validation will be required:

“Similarly, in-depth evaluation of data-driven cytokine response genes will be a next step to purify distinct response signatures.”

*We appreciate the reviewer’s time to investigate Reactome and MSigDB for identifying known cytokine-gene associations, which we believe adds a lot of value. Therefore, we have expanded our analysis to systematically investigate pathway enrichments (**Figure S8 D-I**):*

Figure S8

Fig. S8: Pathway and gene set enrichment analysis of IFNG-related (D, G), IL13-related (E, H) and IL17A-related (F, I) genes.

2. UMIs vs. spots. The authors present their data in Supplemental Table 2. As the authors state “This table also elucidates that presence of cytokines is not distributed equally over the entire dataset, but is specific for the diseases investigated (also shown in Fig. 2N-Q). The mean number of spots per section is 767 ± 293 .” IL-13 is robustly expressed in one atopic dermatitis sample. The next highest expression is half of the most robust at best, but this tissue has 2-3 times the number

of spots, so the overall expression is even less. I am not sure if there is a statistical methods to account for the variability across samples. With apologies, I do not know that Tangram method that the authors report.

We thank the reviewer for this valuable comment. Despite being disease-specific, cytokine expression is unequal distributed over all investigated samples. This reflects the heterogeneity of the investigated diseases observed in clinics, in particular, AD being the most heterogeneous one. We intentionally do not correct for this, since this reflects the disease biology. We have added a comment to the results:

“Generally, respective cytokine transcripts were unequally distributed across all samples, with AD being particularly heterogeneous (Table S2).”

Furthermore, it is not necessary to correct for the size of biopsy, i.e. number of spots per sample, since we anchor on cytokine-positive spots and explore spots in close proximity. Thereby, our results become independent from the size of the biopsy. To make this clear, we have added to the manuscript:

“This enabled us to analyse the impact of detected transcripts on their direct surroundings forming local immune microenvironments and calculate a distinct spatial correlation, independent of sample size and heterogeneous number of cytokines.”

Thanks, we also appreciate the hint that maybe not all readers might be familiar with the Tangram method. Therefore, we briefly introduce Tangram in the results section:

“Given the spot size, a single spot can contain several cells such as immune and epithelial cells that can be identified using deconvolution algorithms such as Tangram generating predictive spatial maps of cell types in a given spot.”

3. Validation by in situ hybridization. The authors have confirmed cytokine expression by in situ hybridization but we don't know if there is variation and whether the differences are significant.

Data on the variation of our performed in situ hybridization is given in Fig. 2G. As indicated, there is quite high variation, especially in the IFNG detection. Differences between groups were not determined as IFNG was solely detected in LP, IL13 in AD and IL17A in Psoriasis (see legend of Fig. 2G):

Fig. 2G) *Quantification of cytokine positive cells per in situ section. Given are IFNG transcript-positive cells in LP (n=5), IL13 transcript-positive cells in AD (n=3) and IL17A transcript-positive cells in psoriasisPso (n=5)*

4. In situ hybridization as compared to the literature. It is appropriate that the authors are referencing the literature.

Thank you for accepting our changes.

5. Responder genes-specificity. The authors have revised the figure to include additional genes.

Thank you for accepting our changes.

6. Validation by scRNA-seq. The authors have included two Figure 4s?

Unfortunately, we could not detect two Figure 4s in the main manuscript. If we have overseen this issue, we are confident that the final publishing process will resolve this.

7 and 8. Leukocytes. I am not clear how NK cells were defined.

NK cells are classically defined as being CD56+ (amongst many other markers), however, they also express CD2 and are therefore also included in the included definition of leukocyte positive spots being CD2, CD3D, CD3E, CD3G, CD247 (CD3Z), or PTPRC (CD45) positive. The reviewer is right that the indicated definition of leukocytes does not immediately draw the attention to NK cells, however, we named them for the sake of completeness.

A minor point:

in revised Figure S3B - the epithelial cell signatures are a minority of the signal in the epidermis, which doesn't make much sense.....and in revised Figure S3D the epithelial component is larger in the dermis than the epidermis.

We agree with the reviewer that the epithelial cell signature may seem odd at first. Therefore, we reanalysed the data and adapted the cell type classification into four major groups, i.e. Lymphoid & Mast cells, APCs, Epidermal non immune, and Dermal non immune as suggested by the original article by Reynolds, Gary et al (48). Thereby, we now observe a more intuitive distribution (Fig. S3B, E, H). Subsequently, we now separately portray the composition of our group of interest, i.e. Lymphoid & Mast cells, thus highlighting the predominance of T-cells (Fig. S3C, F, I). The Tangram algorithm is state-of-the-art, however, it is just a heuristic. It is solely based on a computational prediction, which could also contain false positives and may overestimate certain cell populations, therefore, has to be interpreted with caution. We have added a short comment in discussion:

“To deconvolve spatial spots, we moreover applied the state-of-the-art Tangram algorithm (47), which is a heuristic method to computationally predict the cellular composition of every spot of interest.”

REVIEWERS' COMMENTS

Reviewer #1 (Remarks to the Author):

The authors have satisfied my major concerns - I appreciate their patience with the tedious granularity of review.

The only outstanding issue (where fault may lay with me in failing to identify the explanation) is the use of "only a few cytokine transcripts" in the abstract and "observed that single transcripts of disease-driving cytokines". I think the paper establishes that a likely local amplification of cytokine responder genes occurs as the result of very low (relative to expression of other genes) cytokine expression. On which of the validating data do the authors base their phrasing of absolute cytokine number? It is not on UMIs, correct, which have established issues with sensitivity
<https://genomebiology.biomedcentral.com/articles/10.1186/s13059-018-1438-9>

Reviewer #3 (Remarks to the Author):

The abstract states:

"Leveraging a density-based spatial clustering method, we identified specific responder gene signatures in direct proximity of cytokines, and confirmed that detected cytokine transcripts initiate amplification cascades up to thousands of specific responder transcripts forming localized epidermal clusters. Thus, within the abundant and heterogeneous infiltrates of ncISD, only a few cytokine transcripts drive disease by initiating an inflammatory amplification cascade in their local microenvironment."

The in situ hybridization in the paper and in the literature supports the spatial data indicating that there are few cytokine transcripts and many transcripts for downstream genes, indicating an amplification process which is to be expected for cytokines. However, the statement that "a few cytokine transcripts drive disease by initiating an inflammatory amplification cascade..." is problematic, because cytokine transcripts don't drive disease, it is the encoded protein that drives the amplification cascade. They could fix this overstatement by intercolating protein data from the literature, for example <https://www.ncbi.nlm.nih.gov/pmc/articles/PMC3577967/>- Nicole Wards study.

REVIEWERS' COMMENTS

Reviewer #1 (Remarks to the Author):

The authors have satisfied my major concerns - I appreciate their patience with the tedious granularity of review.

The only outstanding issue (where fault may lay with me in failing to identify the explanation) is the use of "only a few cytokine transcripts" in the abstract and "observed that single transcripts of disease-driving cytokines". I think the paper establishes that a likely local amplification of cytokine responder genes occurs as the result of very low (relative to expression of other genes) cytokine expression. On which of the validating data do the authors base their phrasing of absolute cytokine number? It is not on UMIs, correct, which have established issues with sensitivity

<https://genomebiology.biomedcentral.com/articles/10.1186/s13059-018-1438-9>

Thank you very much for all your comments. They were highly appreciated and improved the manuscript tremendously.

Your concern about UMI counts is reasonable, and the assumptions you are criticizing are based on these counts. It is known that ST, single cell, and bulk RNA-seq data suffer from dropout events: Genes which are measured in a subset of cells but not in other cells and therefore causing an inflation of zero values. However, if our analysis would suffer from dropout events, this would imply

- a varying expression of cytokines between low and high counts – which is not the case as we general observe low numbers of cytokines in all T-cells in general*
- the likelihood of observing these dropout events in all three datasets is quite low because it's a stochastic process and depends also on the number of provided mRNA. Therefore, it shouldn't occur as a pattern in multiple datasets.*

To keep the language used in the abstract, we replaced 'few' by 'low (numbers)' and further added 'translated proteins' to indicate that not transcripts themselves, but their corresponding proteins mediate the effect.

Reviewer #3 (Remarks to the Author):

The abstract states:

"Leveraging a density-based spatial clustering method, we identified specific responder gene signatures in direct proximity of cytokines, and confirmed that detected cytokine transcripts initiate amplification cascades up to thousands of specific responder transcripts forming localized epidermal clusters. Thus, within the abundant and heterogeneous infiltrates of nciSD, only a few cytokine transcripts drive disease by initiating an inflammatory amplification cascade in their local microenvironment."

The in situ hybridization in the paper and in the literature supports the spatial data indicating that there

are few cytokine transcripts and many transcripts for downstream genes, indicating an amplification process which is to be expected for cytokines. However, the statement that "a few cytokine transcripts drive disease by initiating an inflammatory amplification cascade..." is problematic, because cytokine transcripts don't drive disease, it is the encoded protein that drives the amplification cascade. They could fix this overstatement by intercolating protein data from the literature, for example <https://www.ncbi.nlm.nih.gov/pmc/articles/PMC3577967/>- Nicole Wards study.

We thank the reviewer for pointing to this inaccuracy in the abstract. And yes, transcripts do not induce amplifications cascades, but the proteins translated from theses transcripts. We therefore adapted the abstract as follows: 'Thus, within the abundant and heterogeneous infiltrates of nciSD, only a low number of few cytokine transcripts and their translated proteins drive disease by initiating an inflammatory amplification cascade in their local microenvironment.'